# Unified Multimodal Discrete Diffusion

## Abstract

Multimodal generative models that can understand and generate across multiple modalities are dominated by autoregressive (AR) approaches, which process tokens sequentially from left to right, or top to bottom. These models jointly handle images, text, video, and audio for various tasks such as image captioning, question answering, and image generation. While AR models have been highly successful in the text domain, they have been found suboptimal for processing images, videos, and audio due to the high correlation between adjacent tokens which waste inference-time compute by separately predicting each one. In this work, we explore discrete diffusion models as a unified generative formulation in the joint text and image domain, building upon their recent success in the text domain alone. Discrete diffusion models offer several advantages over AR models, including improved control over quality versus diversity of generated samples, the ability to perform joint multimodal inpainting (across both text and image domains), and greater controllability in generation through guidance. Leveraging these benefits, we present the first **Uni**fied Multimodal **Disc**rete Diffusion (UniDisc) model, which is capable of jointly processing text and images for a variety of downstream tasks. We compare UniDisc to multimodal AR models of similar capacity, demonstrating that UniDisc outperforms them in terms of both performance and inference-time compute, enhanced controllability, editability, inpainting, and flexible trade-off of inference time versus generation quality. Additional visualizations are available at `unidisc-diffusion.github.io`

## 1 Introduction

Multimodal generative models—models that can accept and produce a variety of modalities such as text, images, videos, audio—can significantly improve the overall capabilities of an AI system, as these models can (1) leverage information from multiple sources to better understand the context (2) learn from any available data source and (3) respond to a user's request in a flexible manner, thus dynamically generating text, images or audio as required. Although the choice of model architecture—transformers—is currently clear, the optimal generative objective remains unclear.

Current multimodal models are typically trained jointly using (an approximation to) a maximum likelihood objective over multimodal token sequences produced from images, text, and other modalities. AutoRegressive (AR) models primarily quantize tokens from continuous modalities where necessary and optimize the exact likelihood through a series of conditionals; during generation, they use a fixed token order, e.g., left-to-right, top-to-bottom (raster order) for images. They have demonstrated strong performance in both text and image generation, making them the current workhorse for multimodal models. However, generating image tokens autoregressively is slow and wasteful as nearby tokens are highly correlated, and this process results in many unnecessary forward passes through the network Lu et al. (2022); Team et al. (2023); Chameleon (2024). Moreover, AR models are difficult to control Li et al. (2022), cannot inpaint or infill unless explicitly trained, and cannot easily trade-off quality vs compute at inference time.

On the other hand continuous diffusion models—which have been shown to work well for continuous modalities such as images, have fast inference, are highly controllable, and can easily trade-off quality vs compute. These models corrupt data by adding Gaussian noise and are trained to denoise the data, maximizing a lower bound on the likelihood. However these models have been found to perform poorly on text Gulrajani & Hashimoto (2024). Text is inherently discrete, and adding continuous Gaussian noise to text token embeddings does not correspond to meaningful changes in

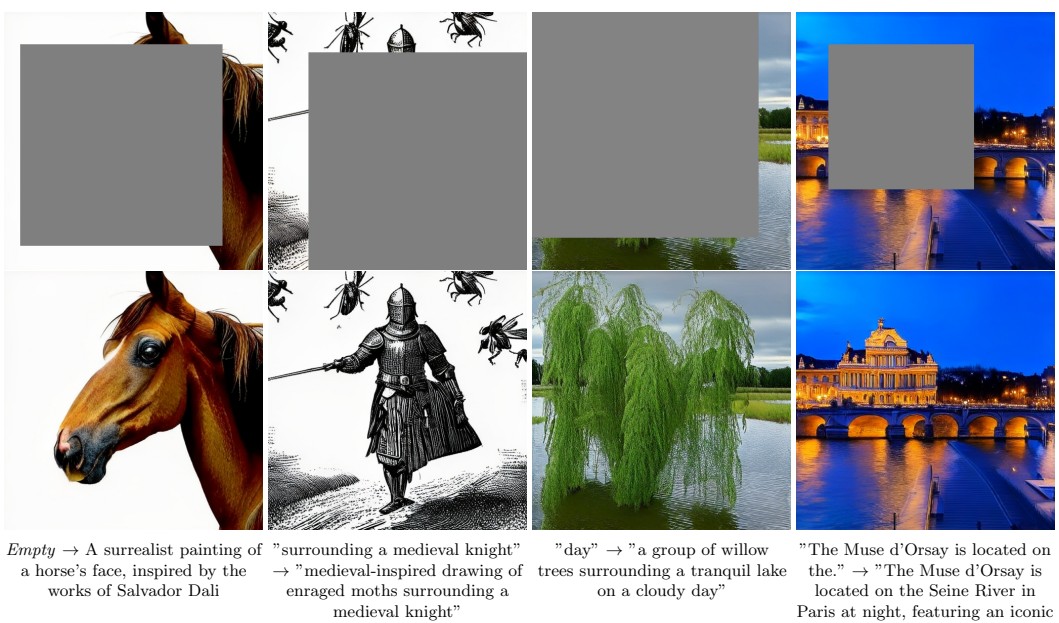

Figure 1: We show UniDisc's ability to jointly inpaint unseen image+text pairs. We do not explicitly optimize for this objective, however it is intrinsic to UniDisc's, unified diffusion objective.

the actual text. These trade-offs between different modeling strategies across modalities raises the question: What is the right unified, generative formulation across text, image, and other modalities?

We present UniDisc, a unified multimodal model based on discrete diffusion. While continuous Gaussian noise is incompatible with discrete data such as text and graphs, UniDisc corrupts data with discrete noise, specifically, randomly masking tokens, and learns to map mask tokens into multimodal tokens during inference. Discrete diffusion through masking noise has been explored separately, both for generating text Austin et al. (2021); Sahoo et al. (2024) and for generating images Chang et al. (2022; 2023). Such explorations have resulted in different noise schedules, transition kernels, and loss functions across the text and image domains. In this paper, we explore a discrete diffusion formulation and its applicability in jointly modeling text and image modalities with a unified set of hyperparameters.

We propose a unified architecture built on top of Transformer that jointly tokenizes text and images, and uses full self-attention to learn to map a masked token sequence to a clean token sequence by sampling from a joint vocabulary of text and image tokens. Unifying different modalities is not straightforward as different modalities have a different number of optimal sampling steps. For instance, we find that text tokens require a lot more sampling steps than the image tokens. We believe this is due to the difference in information density, as image tokens exhibit high correlation across the sequence length. Instead of spending an equal amount of sampling steps (compute) for each modality, we propose a modality-specific caching mechanism. We introduce different noising schedules for each modality, which allows us to spend less sampling steps on images than text, thus significantly improving UniDisc's throughput and latency.

We evaluate UniDisc across multiple image-text datasets, and tasks, such as conditional and unconditional generation, editing and retrieval. We find that UniDisc consistently outperforms its AR counterpart in inference efficiency: at a given inference compute budget, our model achieves generations of higher quality and diversity (Figure 4). Due to classifier-free guidance in conditional generation, UniDisc achieves a much higher FID and CLIP-score than AR (Table 2). UniDisc also showcases stronger discriminative ability than AR, on retrieval tasks, due it's variable sampling steps (Table 3). We further scale UniDisc to a 1.4B parameter model, trained on web-scale image-text datasets. UniDisc exhibits strong joint image-text inpainting abilities that are not possible with prior unified generative models as we show in Figure 1.

Our code, model weights, and dataset will be publicly available upon acceptance. More qualitative visualizations are available at `https://unidisc-diffusion.github.io`.

## 2 RELATED WORK

### 2.1 UNIFIED MULTI-MODAL MODELS

In recent years, unified models for processing multiple modalities have advanced significantly. Models like Flamingo (Alayrac et al., 2022) and PaLM-E (Driess et al., 2023) demonstrate strong few-shot learning capabilities across tasks. LLAVA Liu et al. (2023) enhances LLaMa Touvron et al. (2023) with multimodal fine-tuning, but still uses separate encoders, limiting true unification and image generation. Recent efforts, like Perceiver IO (Jaegle et al., 2021) and Unified-IO (Lu et al., 2022), attempt modality unification but at a smaller scale. The Chameleon project (Chameleon, 2024) scales this up with a 34-billion parameter model trained on image-text data. However these approaches largely focus on autoregressive generation which is inefficient for high-dimensional data.

Relevant to our work, UniD3 Hu et al. (2022) considered discrete diffusion on image and text but made several design decisions that separated each modality, using both absorbing and uniform masking, decoupling the modalities inside the model with separate operations on each. Further we couldn't compare against their model, as were unable to reproduce their reported results using their publicly available code.

### 2.2 DISCRETE DIFFUSION MODELS

Discrete diffusion models have emerged as a promising alternative to continuous diffusion for discrete data types. (Sohl-Dickstein et al.) introduced the first discrete diffusion model over binary variables, (Hoogeboom et al.) extended the noising process to categorical variables, demonstrating its effectiveness on image generation tasks. D3PM Austin et al. (2021) later extended discrete diffusion to a more general set of noising processes, allowing for more flexible noise schedules. Recent work by SEDD (Lou et al.) introduced score entropy, a novel loss function for discrete diffusion models that bridges the gap between continuous and discrete spaces, and more recently, Sahoo et al. (2024); Shi et al. (2024) showed text perplexity competitive with GPT-2. While this approach shows promise for improving discrete diffusion models, these methods were primarily focused on language modeling tasks. Our work extends the application of discrete diffusion to multiple modalities and demonstrates its effectiveness in a unified architecture.

## 3 UNIDISC: UNIFIED DISCRETE DIFFUSION

### 3.1 DIFFUSION MODELS

Diffusion models (Ho et al., 2020; Sohl-Dickstein et al.; Song et al., 2020) are a class of generative models that learn to construct a data distribution by gradually reversing a process that introduces noise into clean data samples. This approach models the transformation of a data sample $x_0$ from a clean state through increasingly noisy states until it reaches a pure noise distribution.

The forward diffusion process is described by a series of transitions where each latent variable $x_t$ at time step $t$ is sampled from a Gaussian distribution as follows:

$$q(x_t|x_0) = \mathcal{N}(x_t; \sqrt{\bar{\alpha}_t}x_0, (1 - \bar{\alpha}_t)I)$$

Here, $\bar{\alpha}_t = \prod_{s=0}^{t} \alpha_s$ represents the cumulative product of noise levels, making $x_t$ increasingly distant from $x_0$ as $t$ increases. The variable $x_t$ represents the noisy version of $x_0$ at time $t$, modeled to progressively approximate Gaussian noise as $t$ approaches the final time step.

The reverse diffusion process then aims to reconstruct the original data by progressively denoising these samples. This involves learning the reverse transitions, with the goal to train the model $p_\theta(x_{t-1}|x_t)$ to approximate the true reverse process and effectively recover the original data point $x_0$ from the noisy samples.

Given $T$ timesteps of diffusion, the loss using the Evidence Lower Bound (ELBO) for the diffusion process equals[1]:

$$\mathcal{L}_{\text{diff}} = \underbrace{-\mathbb{E}_{q(x_1|x_0)}\left[\log p_\theta(x_0|x_1)\right]}_{\text{reconstruction term}} + \underbrace{\sum_{t=2}^{T}\mathbb{E}_{q(x_t|x_0)}\left[D_{KL}(q(x_{t-1}|x_t,x_0)\|p_\theta(x_{t-1}|x_t))\right]}_{\text{denoising matching term}} \quad (1)$$

## 3.2 Discrete Diffusion Models

Building on the foundations of continuous diffusion models, discrete diffusion models adapt these concepts to structures that are inherently discrete. Unlike their continuous counterparts that model transitions of $x_t$ given $x_{t-1}$ with Gaussian distributions, discrete models define transitions using categorical distributions. The forward process for discrete models is thus characterized as:

$$q(x_t|x_0) = \text{Cat}(x_t; x_0 \cdot \bar{Q}_t) \quad (2)$$

Here, $\bar{Q}_t = \prod_{t=0}^{t=t} Q_t$ which represents the cumuliative transition matrix at each discrete time step $t$, where $Q_t$ is a transition matrix $[Q_t]_{ij} = q(x_t = j \mid x_{t-1} = i)$ dictating the probabilities of moving from one discrete state $x_{t-1}$ to another $x_t$, and $x_0$ is a one-hot vector of the input data sample. D3PM (Austin et al., 2021) generalizes this framework over various transition matrices, the popular ones mainly include uniform and absorbing transition matrix. In UniDisc, we use the absorbing transition matrix as emperically it has been found to work the best across text and images Austin et al. (2021); Lou et al. (2024). Absorbing transition matrix requires having an absorbing state namely the [MASK] token. The matrix is represented as $Q_t = \alpha_t I + (1 - \alpha_t)\mathbb{1}e_m^T$, where $\mathbb{1}$ is a column vector of ones and $e_m$ is a one-hot vector with one on the mask state $m$. This ends up being a matrix with all zeros except $i = j \neq m$ is $\alpha$ and $j = m, i \neq m$ is $1 - \alpha$ and $i = j = m$ is 1.

Intuitively this means that during the forward transition, the probability of an input token $x_0$ to stay the same is $\alpha$, the probability of it being masked is $1 - \alpha$, and the probability of a masked token to be unmasked is 0.

Given the forward diffusion in equation 2, (Sohl-Dickstein et al.) uses the same objective function as Equation 1 to optimize their model, where $q(x_{t-1}|x_t)$ ends up being a Bernoulli distribution instead of a Gaussian distribution. MDLM Sahoo et al. (2024) simplifies this objective function, by considering continuous time-diffusion and applying loss only on the masked tokens. The final loss simply ends up being a re-weighted masked generative modeling loss:

$$\mathcal{L}_{\text{diff}} = \mathbb{E}_{t\sim\mathcal{U}(0,1),q(x_t|x)}\left[\frac{\alpha_t'}{1-\alpha_t}\log p_\theta(x_0 \mid x_t)\right] \quad (3)$$

where $\alpha_t' = \alpha_t - \alpha_{t-1}$, and $\alpha_t$ is the probability of the token not being masked. MaskGIT (Chang et al., 2022) and Muse, state-of-the-art masked image generative model use the same loss as Eq 3, except there is no re-weighting term and the time is discrete time instead of continuous time. The noising schedule $\alpha_t$ is also different, while language discrete diffusion models such as (Austin et al., 2021; Sahoo et al., 2024) use a linear-time schedule, MaskGIT and Muse (Chang et al., 2022; 2023) use a cosine schedule. We ablate these different design choices in our experiment Section.

## 3.3 Unified Training via UniDisc

We train a Bidirectional Decoder-Only Transformer architecture of Vaswani et al. (2017) with Rope embeddings Su et al. (2023). We use 2D RoPE (Liu et al., 2024) for all image tokens and 1D RoPE for text tokens, and add learned modality-specific embeddings to each token. This allows our model both flexibility in resolution at inference, and the ability to use compute effectively by performing the majority of training at a lower resolution. We use the same objective function as Equation 3, except for us $x_0 \leftarrow [x_0^{img}, x_0^{txt}]$

Classifier-Free guidance (CFG) (Ho & Salimans, 2022) has been used in continuous diffusion models to trade-off between quality and diversity of generation. We apply this idea to discrete diffusion,

---

[1]We skip the prior matching term from the loss as it is assumed to be zero

with a probability of 0.1 we set all the tokens of a random modality to be mask tokens, this allows UniDisc to learn unconditional likelihood for image and text modality. During inference we use CFG for conditional generation (image-to-text or text-to-image) to trade-off between quality and diversity of generation as shown in Figure 4 and Figure 10

Empirically, we find that the number of sampling steps required for Image decoding is much lower than for text decoding. As can be seen in Figure 4 (c) and (d), FID saturates within 32 sampling steps for images, while it takes 400 sampling steps for the text generative perplexity to saturate. This creates an issue for unified modeling as the number of sampling steps is bottlenecked by the maximum number of sampling steps of any input modality. To resolve this issue, we propose KV-caching the tokens of the faster modality, which in this case is the image. This however is not feasible with current training scheme, as the text tokens never encounter, the image tokens from the previous denoising steps.

To fix this we propose to have different time schedules for each modality, specifically a slower time-schedule for text and a faster one for images. To implement these schedules, we consider $N_{min}$ and $N_{max}$, which represent the min and max number of timesteps one might consider during inference. We also consider $K$, which represents the text-to-image inference ratio, i.e how many times is image inference faster than text inference. Empirically, we find this number to be roughly 10. During training, we first randomly sample text timesteps from a uniform distribution, $t_{txt} \sim \mathcal{U}(0, 1)$ and set image timesteps by randomly offsetting the text timesteps with $\delta t_i$, specifically $t_{img} \sim \mathcal{U}(t_{txt}, t_{txt} - \delta t_i)$. This ensures that the image timestep only moves behind the text timestep by a maximum of $\delta t_i$. We randomly set $\delta t_i$ to $\delta t_i \sim \mathcal{U}(\frac{K}{N_{max}}, \frac{K}{N_{min}})$. This modality-specific timestep schedule ensures UniDisc can handle image token KV caching during inference.

To improve training stability, we use Query-Key Normalization Wortsman et al. (2023) and use RMSNorm Zhang & Sennrich (2019) for all other norms. We use Sandwich Normalization—normalization before and after each FFN, as we found this helps control activations in deeper layers as previously reported in Ding et al. (2021); Zhuo et al. (2024).

To further improve the convergence speed of discrete diffusion we analyze the noising schedule and find that linear schedule in (Sahoo et al., 2024; Austin et al., 2021) results in excessively high weighting for early timesteps, impairing the convergence speed. Following Min-SNR trick in continuous diffusion Hang et al. (2023), we limit the minimum weighting to 5. We provide the pseudo-code for training procedure in Algorithm A.1.

### 3.4 Unified Sampling via UniDisc

Sampling in masked discrete diffusion, involves mapping a set of masked tokens $m$ to a set of visible tokens $x_0$ using $T$ timesteps of denoising. A variety of sampling strategies have been previously proposed (Sohl-Dickstein et al.; Austin et al., 2021; Zheng et al., 2024; Chang et al., 2022; Sahoo et al., 2024; Lou et al., 2024) for masked discrete diffusion. MaskGIT (Chang et al., 2022) proposes a confidence-based sampling, where they decode the most confident tokens at each step of denoising. D3PM (Austin et al., 2021) and MLDM (Sahoo et al., 2024) uses a sampling mechanism similar to (Ho et al., 2020) except applied to bernoulli distribution, which we refer to as DDPM sampling. This results in a random set of tokens being decoded, instead of the

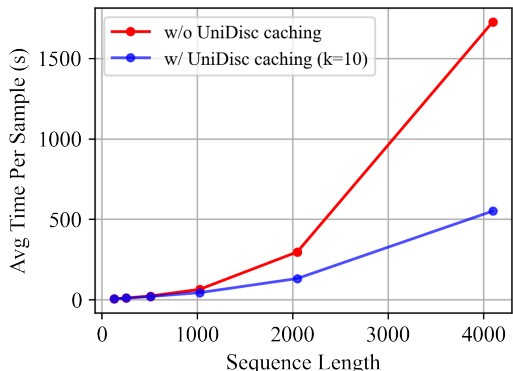

Figure 2: Latency vs Seq Length for our caching approach - image-to-text tokens ratio = 1/4. We empirically find $k = 10$ from Figure 4 based on the saturation steps for image to text.

most confident ones as in MaskGIT. We ablate these sampling strategies in Figure 4, and find the confidence-based sampling proposed in MaskGIT to work the best for unified modeling.

We also build on top of MaskGIT sampling and add modality-specific caching and nucleus sampling on the logits. In modality-specific caching, we cache the image token keys/values, every $k$ steps. We provide the pseudo-code of our sampling in Algorithm 2. In Figure 2, we find that KV caching the

image tokens results in much lower latency at higher sequence length or batch size. Our algorithm along with our MaskGIT implementation is available in A.2.

# 4 EXPERIMENTS

We compare UniDisc against an autoregressive (AR) baseline across various tasks, metrics and datasets. We use the same architecture and hyper-parameters, and data, only differing in the attention mask and respective loss functions. For our autoregressive baseline we use a standard language model architecture from Chameleon (Chameleon, 2024)—that is a decoder-only transformer with causal attention and rotary positional embeddings. To enable classifier-free guidance, we dropout modalities with $10\%$ probability during training. For UniDisc, we dropout both modalities and for the AR baseline we dropout only the first modality in the input sequence as in (Liu et al., 2024).

Our experiments aim to answer the following questions:

1. How does UniDisc compares against AR models in unconditional and conditional multi-modal generation of image/text pairs?
2. How effective is classifier-free guidance in conditional generation for AR models and for UniDisc?
3. How does UniDisc compare against AR models in terms of training efficiency with varying the ratio of image-text tokens?
4. How do various sampling strategies for UniDisc affect its generation results and inference speed?
5. How does UniDisc compare against AR models across image-language reasoning tasks?
6. How do various design choices of UniDisc contribute to its performance?

Lastly, we show that we can successfully scale UniDisc, to a 1.4 billion parameter model, trained on 500B tokens. We qualitatively evaluate this model, to demonstrate its capabilities.

**Datasets:** In Section 4.1, 4.2 and 4.3, we conduct experiments with different train and validations sets. Our training set includes DataComp1B (Gadre et al., 2024), CC12M (Changpinyo et al., 2021), CLEVR-math (Lindström & Abraham, 2022) and CLEVR-Ref (Liu et al., 2019). Our evaluation datasets include a held-out validation set of DataComp1B and CC12M, along with Flickr, MS-COCO30k (Chen et al., 2015)(Plummer et al., 2016) and Winoground (Thrush et al., 2022).

## 4.1 UNCONDITIONAL & CONDITIONAL MULTIMODAL GENERATION

We evaluate UniDisc and AR models in unconditional and conditional generation tasks.

**Evaluation metrics:** We consider the following three evaluation metrics most commonly used in previous works: i) **Joint perplexity** indicates a model's ability to fit to different validation sets. Note that this metric is jointly calculated across image-text tokens. The perplexity values from the autoregressive Chameleon baseline are exact likelihoods, the values from UniDisc are upper bounds. While perplexity is a good metric for assessing the fitting ability of a model, it cannot be used to evaluate its generation ability. ii) **Fréchet inception distance (FID)** Heusel et al. (2017) is a popular metric in image-generation to quantify the quality and diversity of image generation. We use this to quantify the quality of the generated images from

| | CC12M | DataComp | Flickr | MS-COCO |
|---|---|---|---|---|
| **Image + Text Perplexity** | | | | |
| Chameleon | 541.2 | 156.8 | 1254.9 | 1128.3 |
| UniDisc | 494.5 | 154.8 | 1115.0 | 982.2 |
| **Image - FID** | | | | |
| Chameleon | **30.5** | **20.49** | **75.70** | **70.67** |
| UniDisc | 35.78 | 22.97 | 88.88 | 77.43 |
| **Text - CLIP** | | | | |
| Chameleon | 23.70 | **26.08** | 23.70 | 23.64 |
| UniDisc | **25.01** | 25.98 | **24.92** | **25.01** |

Table 1: Unconditional multimodal generation results for UniDisc and AR baseline at 115M parameters - both models perform similarly.

diversity of image generation. We use this to quantify the quality of the generated images from

unconditional sampling. **iii) CLIP-score** is used for calculating image-text coherence. While we could not find an equivalent FID metric for text, we use CLIP score to evaluate generated image-text coherence, conditioning our model on an input image. For unconditional generations in Table 1, we compute the CLIP score between our generated image and generated text. For conditional generations in Table 2, we compute the CLIP score by conditioning our model on ground-truth images.

**Experimental details:** For unconditional and conditional results in Table 1, Table 2, Figure 3 we use a dataset of 11B tokens comprising 30M images from DataComp1B (Gadre et al., 2024) and CC12M (Changpinyo et al., 2021) as our training set, with a fraction of $20\%$ text tokens and $80\%$ image tokens after excluding pad tokens. For faster convergence, we train only on DataComp1B for results in Figure 4 and Figure 5. We tokenize the image and text tokens using seperate tokenizers. We use lookup-free quantization (LFQ) from (Yu et al., 2023; Luo et al., 2024) for as our image tokenizer, and use the tokenizer from (Touvron et al., 2023) as our text tokenizer. We use an image resolution of $256 \times 256$, and a downsampling ratio of 16, resulting in a sequence length of 384 with 256 with image tokens and 128 text tokens. Note that we use the same tokenizers for all the baselines, ensuring fair comparisions. We train UniDisc for 300 L40S GPU hours, following Figure 3 we train the autoregressive model for a proportionate amount of time such that it achieves the same validation loss. Our model comprises 115M/340M non-embedding parameters and we use a batch size of 512, a learning rate of $3e-4$, and LR decay of 0.05, following (Sun et al., 2024).

We show unconditional image-text generation results in Table 1, and conditional generation results in Table 2. For conditional generation, we condition on an image to generate the corresponding language description, and vice versa, condition on the language description to generate the corresponding image. Although unconditional generation is useful, almost all use-cases of current generative models concern conditional generation, i.e the user provides a prompt and the model responds accordingly. As seen from the Tables, while UniDisc significantly outperforms AR in conditional generation while performing equally well in conditional generation. We attribute this performance gap to classifier-free guidance (CFG). As can be seen in Table 2, w/o CFG results for AR and UniDisc are similar. By adding CFG, while the AR results do not improve significantly, there is a significant improvement for UniDisc.

|  | CC12M | DataComp | Flickr | COCO |
|---|---|---|---|---|
| **Text to Image - FID** | | | | |
| Chameleon 115M w/o CFG | 26.32 | 20.49 | 46.13 | 56.46 |
| Chameleon 340M w/o CFG | 20.75 | 18.53 | 36.24 | 42.41 |
| Chameleon 115M w/ CFG (0.5) | 22.10 | 16.68 | 46.06 | 47.58 |
| Chameleon 340M w/ CFG (0.5) | 20.22 | 13.55 | 32.74 | 30.62 |
| UniDisc 115M w/o CFG | 27.22 | 21.26 | 43.46 | 54.21 |
| UniDisc 340M w/o CFG | 19.28 | 14.59 | 34.37 | 37.73 |
| UniDisc 115M w/ CFG (1.5) | **13.21** | **12.00** | **33.79** | **31.94** |
| UniDisc 340M w/ CFG (1.5) | **13.11** | **11.55** | **26.83** | **23.77** |
| **Image to Text - CLIP** | | | | |
| Chameleon 115M w/o CFG | 22.08 | 26.01 | 22.50 | 23.02 |
| Chameleon 340M w/o CFG | 22.53 | 26.68 | 23.51 | 24.46 |
| Chameleon 115M w/ CFG (0.5) | 22.93 | 27.30 | 23.38 | 24.03 |
| Chameleon 340M w/ CFG (0.5) | 23.65 | 27.70 | 24.95 | 25.99 |
| UniDisc 115M w/o CFG | 21.75 | 25.98 | 22.44 | 22.88 |
| UniDisc 340M w/o CFG | 22.18 | 26.86 | 23.18 | 24.44 |
| UniDisc 115M w/ CFG (1.5) | **24.54** | **29.65** | **25.42** | **26.24** |
| UniDisc 340M w/ CFG (1.5) | **24.77** | **30.01** | **26.63** | **27.82** |

Table 2: Conditional generation results for UniDisc and AR baseline. Our model significantly outperforms the AR model when classifier free guidance is used.

The iterative generation process of diffusion makes it easy to blend conditional and unconditional predictions to guide the output. Autoregressive models, on the other hand, generate data sequentially in a fixed order, without any iterative refinement, which makes it difficult to mix in unconditional predictions to guide generation.

## 4.2 TRAINING COMPUTE AND & INFERENCE SPEED

With the ever growing scale of contemporary generative models, an important aspect of their performance is their compute efficiency, divided into training and inference efficiency.

Training efficiency measures the compute cost for a model to achieve a certain negative log likelihood (NLL) of the data distribution (Kaplan et al., 2020). Inference efficiency measures the latency or throughput with which a generative model generates samples. While there have been many works measuring the training scaling laws of autoregressive models (Kaplan et al., 2020; Hoffmann et al.,

2022), there has been almost no work measuring the training efficiency of discrete diffusion models. The closest work is that of Gulrajani & Hashimoto (2024), which compares the scaling efficiency of continuous diffusion models and AR models on text data. They find continuous diffusion models to be about 64x more training inefficient than AR models.

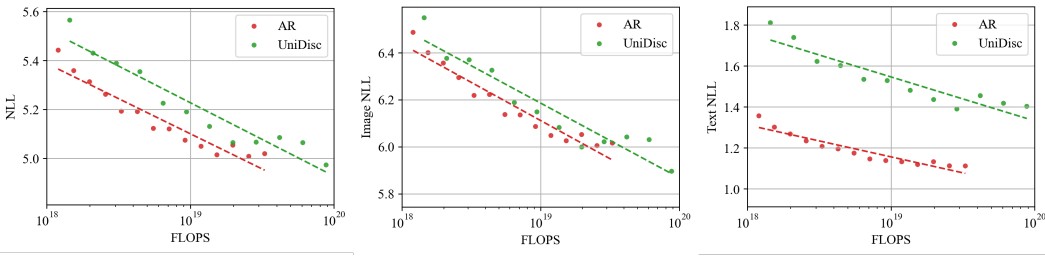

Figure 3: Training Efficiency of UniDisc vs AR. X-axis is training time or FLOP/s. Y-axis is NLL

We compare the training efficiency of UniDisc and our AR baseline in Figure 3 (leftmost column) and find that the rough training-inefficiency factor for discrete diffusion to that of AR for unified training is about 8, which means one needs to train UniDisc 8x longer to achieve the same loss. Interestingly we find that this factor is much less for image-only likelihood, and significantly worse if we consider the text-only likelihood.

While training efficiency is important, inference efficiency is equally—if not more—important as we deploy these models at wide scale. Thus, we compare the inference efficiency of UniDisc and our AR baseline in Figure 4 (a), (c) and (d). In (a), we measure the joint generative perplexity using chameleon, In (c) we measure the Image FID and in (d) we measure the Text Perplexity. While it might appear from (a) and (d) that AR does better than UniDisc. In Figure 4 (b), we find that UniDisc has far higher entropy at a given perplexity.

We note that solely looking at the generative perplexity is not sufficient, as it has been previously found (Zheng et al., 2024) that very low perplexity can be achieved by repeating the same tokens, which we find often happens with AR w/nucleus sampling and low temperature. Therefore Generative Perplexity + Entropy is a better indication of the quality of generation results.

### 4.3 IMAGE-TEXT RETRIEVAL

Recently, several works (Li et al., 2023; Jaini et al., 2024; Prabhudesai et al., 2023) have shown that popular generative models can be strong discriminative models. Additionally (Rambhatla & Misra, 2023) show the discriminative ability of a generative model can be a good metric to assess its generation performance.

In this section, we compare the discriminative capabilities of AR models and UniDisc. We evaluate on Winoground (Thrush et al., 2022) and a held-out DataComp1B validation set (Gadre et al., 2024), using 18M

|  | Clevr-VQA | Clevr-Ret | Datacomp | Winoground |
|---|---|---|---|---|
| **Text Retrieval** | | | | |
| Chameleon | 0.60 | 0.81 | 0.85 | 0.24 |
| UniDisc | **0.63** | **0.94** | 0.85 | **0.31** |
| **Image Retrieval** | | | | |
| Chameleon | N/A | 0.06 | **0.96** | 0.25 |
| UniDisc | N/A | **0.25** | 0.95 | **0.27** |
| **Joint Retrieval** | | | | |
| Chameleon | N/A | 0.06 | 0.17 | 0.06 |
| UniDisc | N/A | **0.5** | **0.64** | **0.20** |

Table 3: Image-Text Reasoning measured by QA and retrieval accuracy across datasets.

text/image pairs from DataComp1B as our training set. To enable text retrieval during inference for the AR model, we train with flipping the order of modalities, putting the image first 20% of the time, following (Zhou et al., 2024). We find that this improves the retrieval for the AR model. All other hyperparameters follow those in Section 4.1.

For evaluations on CLEVR-VQA and CLEVR-Ref (Liu et al., 2019) we use their respective train-val splits. Note that for CLEVR-VQA and CLEVR-Ref, we do not follow the training scaling factor found in Figure 3, we instead train both the models until convergence, i.e multiple epochs. The small

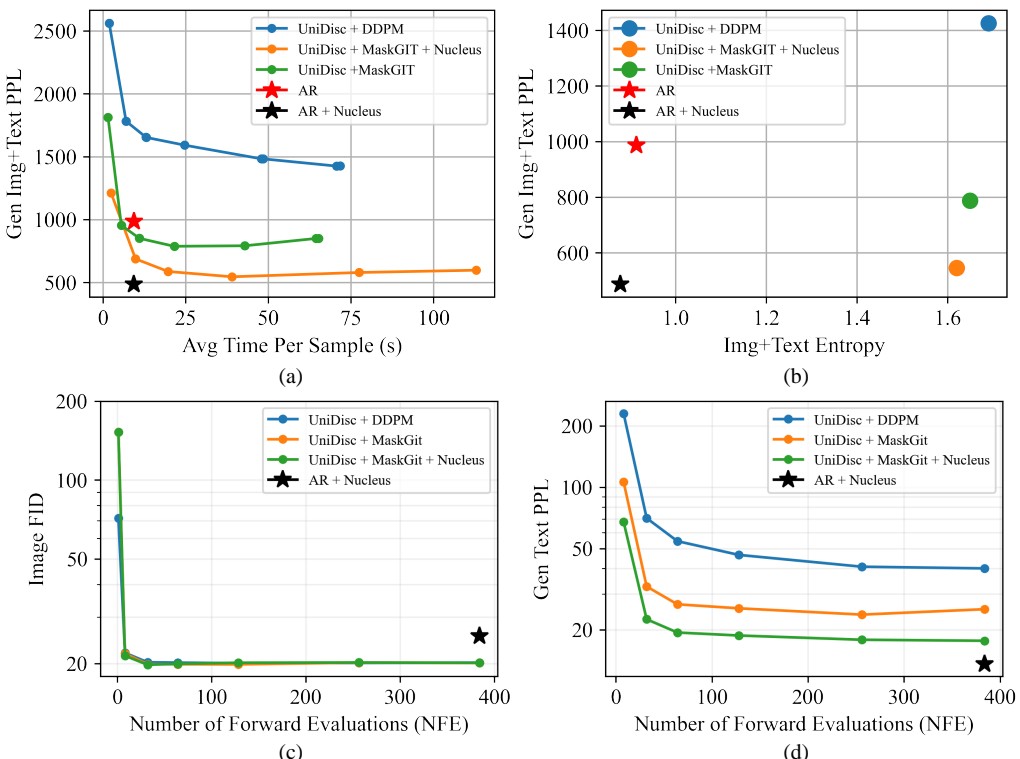

Figure 4: Inference Comparisons for UniDisc and AR baseline: (a) Chameleon (Text+Image) Perplexity vs Time - we perform similar to best AR method, (b) Chameleon Perplexity vs Entropy - UniDisc has high diversity and low perplexity, while AR has significantly lower diversity, (c) Image FID vs NFE, showing image generation saturates quickly with NFE ($\approx$ 32), (d) GPT2 Generative Text Perplexity vs NFE showing text generation benefits from more sampling steps (diminishing).

size of these datasets makes it possible to train until convergence. For CLEVR Images, we find that none of the existing tokenizers work well, so we fine-tune our own tokenizer on CLEVR images. We use images of $128 \times 128$ resolution, with a total sequence length of 320 (256 image tokens and 64 text tokens). For text, we use a standard BertTokenizer (Devlin et al., 2019).

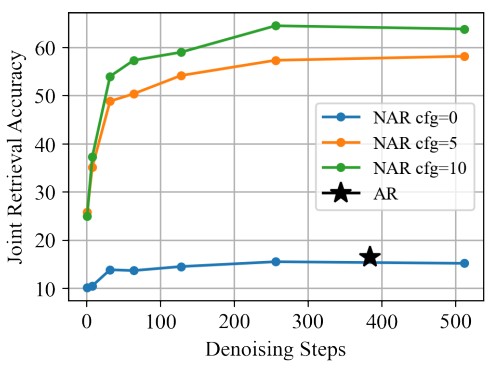

Figure 5: Joint Retrieval Accuracy on DataComp1B - We significantly outperform AR retrieving one correct pair among 15 incorrect pairs.

In Table 3, we report the image retrieval, text retrieval and joint retrieval accuracy for AR and UniDisc. For image retrieval, the model is given a text caption paired with 16 images, out of which only one image is correctly paired and the rest are random. The goal is to accurately classify the correct image. To evaluate the model's retrieval accuracy we check if the correct image has the highest $p(x^{img}|x^{txt})$ among all other images. We do the same for text retrieval, where we check $p(x^{txt}|x^{img})$. For joint retrieval, only a single pair has the correct mapping, and every other pair has a random image and text. We check if the correct pair has the highest joint probability $p(x_{img}, x_{txt})$

We find that UniDisc significantly outperforms the AR model on all retrieval tasks. To further investigate this, we measure the joint retrieval accuracy across denoising steps & CFG values in Figure 5. We find CFG and the number of denoising steps to play a large role in UniDisc's retrieval accuracy. While the number of denoising steps in an AR model is fixed to the sequence length, the denoising steps for UniDisc can be much higher.

## 4.4 IMAGE-TEXT INPAINTING

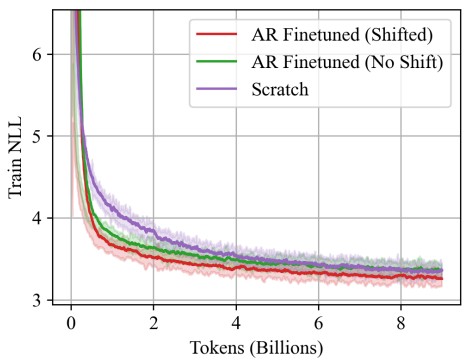

Figure 6: Fine-tuning a pretrained 270M parameter AR model on LM1B

The ability to jointly inpaint in image and text space can be very useful for personalized content creation. Currently, none of the popular generative models have this capability. This is because most unified multimodal generative models are either autoregressive (Team, 2024) or use mixed modeling (Zhou et al., 2024), which prevents them from jointly inpainting in text-image space. In Figures 1, 11, and 12, we show the joint inpainting capabilities of UniDisc. As can be seen, without any fine-tuning, UniDisc, can effortlessly inpaint across modalities.

## 4.5 FINE-TUNING AN AUTOREGRESSIVE MODEL FOR DISCRETE DIFFUSION

Since we already have a plethora of large-scale AR models (Touvron et al., 2023; Chameleon, 2024), it would be useful to have the ability to fine-tune them for discrete diffusion objective. While the naive method for fine-tuning would be to change the objective function to discrete diffusion while using AR's pre-trained weights. We find that a better idea is to left-shift the output targets of the diffusion objective such that instead of having the masked token predict its respective visible token, we have the token before the masked token predict it. Thus it more closely matching AR's next-token prediction objective. In Figure 6 we show that this strategy works well and we can effectively fine-tune a pretrained autoregressive language model using discrete diffusion loss. We demonstrate this result on a 270M parameter language model (Mehta et al., 2024), OpenELM, which is trained with an AR objective. We compare against training from scratch and training AR without the shift. We find shifting strategy converges faster.

## 4.6 SCALING UNIDISC

We show that UniDisc scales well across parameters and dataset size. We train a 1.4B parameter model with web-scale data. Our 1.4B model is trained in two stages, with a low-resolution pre-training stage and a second high-resolution fine-tuning stage. Our first-stage consists of 250M image/caption pairs at 256x256 resolution. We curate our dataset from several sources, with 200M open-web images from (Gadre et al., 2024), which were re-captioned by a VLM to create higher-quality descriptions by (Li et al., 2024b). We also add a set of smaller, PixelProse (Singla et al., 2024), SegmentAnything (Chen et al., 2023), and JourneyDB (Sun et al., 2023). In addition, we construct a high-quality, custom dataset of 18M synthetic images, following findings by (Zhuo et al., 2024; Sehwag et al., 2024) on the importance of image/caption alignment for image generation. We construct our dataset by prompting an LLM to augment a set of 250K human prompts and use (Esser et al., 2024) for generations. In both stages, we account for dataset imbalance and sample more from higher-quality sources. Finally, we fine-tune our model in a second stage, interpolating the RoPE 2D embeddings to train at 512x512 on 30M image/caption pairs.

Additional ablations of smaller models are available in A.3. The training curve of our 1.4B model along with additional qualitative results with different masking strategies and varying classifier guidance are available in A.5 and A.6 respectively.

## 5 CONCLUSION

In this paper, we introduced UniDisc, the first large-scale unified multimodal discrete diffusion model capable of generating and editing both images and text. By leveraging discrete diffusion processes, we showed that UniDisc surpasses autoregressive models in both inference efficiency and quality. Our model unifies various design choices in discrete diffusion space, across modalities, through extensive ablations and analysis. We hope that our work inspires future research in this direction.

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

# A APPENDIX

## A.1 UniDisc Training

We describe the detailed algorithm for unified discrete diffusion training on image and text below in Algorithm 1.

---

**Algorithm 1** UniDisc Training

---

1: **Require:** Training data $x$
2: **Require:** Noising Schedule $\alpha_t$ i.e Linear or Cosine
3: **Require:** Unconditional probability $p_{uncond}$
4: **Optional Inputs:** Text-image inference ratio: $K$,
5: **Optional Inputs:** Min & Max Sampling Steps: $N_{min}$ & $N_{max}$
6: **Initialize:** Model parameters $\theta$
7: **repeat**
8:     $[x_0^{img}, x_0^{txt}] = x_0 \sim p(x, c)$          $\triangleright$ Sample image, text data
9:     $t_{txt} \sim \mathcal{U}(0, 1)$          $\triangleright$ Sample random timestep
10:     **If** UniDiscSampling:
11:         $\delta t_i \sim \mathcal{U}(\frac{N_{min}}{K}, \frac{N_{max}}{K})$
12:         $t_{img} \sim \mathcal{U}(t_{txt}, t_{txt} - \delta t_i)$
13:     **Else**:
14:         $t_{img} \leftarrow t_{txt}$
15:
16:     $x_{t_v}^v \sim q(x_{t_v}^v \mid x_0^v) = \alpha_{t_v} x_0 + (1 - \alpha_{t_v}) e_m$    for $v \in \{\text{img}, \text{txt}\}$      $\triangleright$ mask all tokens
17:
18:     **With probability** $p_{uncond}$          $\triangleright$ For Classifier-Free Guidance:
19:       **If** rand() $< 0.5$:          $\triangleright$ Randomly set one of the modalities to mask tokens
20:         $x_t^{img} \leftarrow m$
21:       **Else**:
22:         $x_t^{txt} \leftarrow m$
23:     $x_0^{pred} = p_\theta([x_t^{img}, x_t^{txt}])$          $\triangleright$ Estimate model prediction from masked sequence
24:     Compute loss as: $\mathcal{L}_{\text{diff}} = \frac{\alpha_t'}{1 - \alpha_t} \log\langle x_0^{pred}, x_0 \rangle$      $\triangleright$ Loss function over the logits of inputs
25:     Perform gradient step on $\mathcal{L}$ to update $\theta$
26: **until** converged

---

## A.2 SAMPLING ALGORITHMS

Here we describe the implementations of UniDisc's sampling algorithm and MaskGIT (Chang et al., 2022).

---

**Algorithm 2** UniDisc Sampling

---

1: **Initialize:** $x_T \leftarrow [m, m, \ldots, m]$          ▷ All tokens are masked
2: **Initialize:** $KV_{img} \leftarrow \emptyset$       ▷ Initialize KV-cache for image tokens to null set
3: **Require:** Sampling steps $T$, Text-image inference ratio $K$
4: **Require:** Num Tokens to Unmask: $f(t)$. We set $f(t)$ as $\frac{1-\alpha_t}{\sum_{t=1}^{T} 1-\alpha_t}$
5: **for** $t = T$ **down to** 1 **do**
6:    **If** $t$ modulo $K == 0$:
7:      $p_{x_0}, KV_{img} \leftarrow p_\theta(x_0 \mid x_t)$    ▷ Model predictions and KV image tokens for caching.
8:    **Else**:
9:      $p_{x_0}^{txt} \leftarrow p_\theta(x_0^{txt} \mid x_t^{txt}, KV_{img})$   ▷ Use cached image tokens from previous timesteps
10:      $p_{x_0} \leftarrow p_{x_0}^{txt}$       ▷ Fix image tokens, Only sample text tokens
11:    $p_{x_0}^{(k)} \leftarrow \text{Top}_k(p_{x_0})$         ▷ Top-$k$ filtering on logits
12:    $p_{x_0}^{(k)} \leftarrow \frac{p_{x_0}^{(k)}}{\tau(t)}$          ▷ Apply temperature annealing
13:    Sample $x_{\text{new}} \sim \text{Categorical}(p_{x_0}^{(k)})$      ▷ Sample new tokens
14:    $M \leftarrow \lfloor f(t) \times N \rfloor$      ▷ Determine number of tokens to unmask
15:    Select $M$ most confident tokens based on $p_{x_0}^{(k)}$
16:    Update $x_{t-1}[i] \leftarrow x_{\text{new}}[i]$   $\forall i \in$ selected positions
17:    **Keep** previously unmasked tokens unchanged
18: **end for**

---

**Algorithm 3** MaskGIT Sampling

---

1: **Initialize:** $x_T \leftarrow [m, m, \ldots, m]$          ▷ All tokens are masked
2: **Require:** Sampling steps $T$
3: **Require:** Num Tokens to Unmask: $f(t)$. We set $f(t)$ as $\frac{1-\alpha_t}{\sum_{t=1}^{T} 1-\alpha_t}$
4: **for** $t = T$ **down to** 1 **do**
5:    $p_{x_0} \leftarrow p_\theta(x_0 \mid x_t)$          ▷ Model prediction
6:    $p_{x_0}^{(p)} \leftarrow \text{Top}_p(p_{x_0})$       ▷ Top-$p$ (Nucleus) sampling on logits
7:    $p_{x_0}^{(k)} \leftarrow \frac{p_{x_0}^{(k)}}{\tau(t)}$          ▷ Apply temperature annealing
8:    Sample $x_{\text{new}} \sim \text{Categorical}(p_{x_0}^{(k)})$      ▷ Sample new tokens
9:    $M \leftarrow \lfloor f(t) \times N \rfloor$      ▷ Determine number of tokens to unmask
10:    Select $M$ most confident tokens based on $p_{x_0}^{(k)}$
11:    Update $x_{t-1}[i] \leftarrow x_{\text{new}}[i]$   $\forall i \in$ selected positions
12:    **Keep** previously unmasked tokens unchanged
13: **end for**

---

## A.3 ABLATIONS

We validate our design choices by running small-scale experiments on a subset of our primary dataset, taking 18M image/caption pairs on DataComp1B. We train on lower-resolution images at $128 \times 128$ and obtain a 1:1 ratio of text to image tokens, with 64 text and 64 image tokens for a total sequence length of 128, with all other hyperparameters the same as in our primary experiments.

We examine the influence of several design choices for our model in Table 4 and reach several conclusions. First, architecture changes to improve training stability—namely adding QK Normalization and using RMSNorm instead of LayerNorm—do not substantially affect convergence in this setting.

Another natural design choice is to parameterize the model such that we provide the modality of a given token to the model. With this relaxation we can drastically reduce the output space and, in theory, simplify the objective for our model. However, we find that this reparameterization only marginally reduces our overall perplexity, even at this smaller-scale. We hypothesize that the modality-specific embeddings added to each token allows the model to learn the correct output space with minimal added parameters.

| | DataComp1B Validation PPL |
|---|---|
| UniDisc | 93.8 |
| w/o QK Norm | 92.7 |
| w/ Zero-linear init | 93.8 |
| w/o RMSNorm | 93.8 |
| w/o -inf for invalid tokens | 94.7 |
| w/o Softmin SNR | 109.6 |
| None | 111.2 |

Table 4: Ablation w/115M parameter model on QK Norm, zero initialization of linear layers, RMSNorm, setting invalid tokens to $-\infty$ during training and generation and Softmin SNR.

| | DataComp1B Validation FID |
|---|---|
| UniDisc | 11.4 |
| w/cosine noising schedule | 11.5 |
| w/o CE loss weighting | 11.35 |
| w/discrete time (T=1000) | 13.8 |

Table 5: Ablation w/115M parameter model on different objective level decisions such as noising schedule, loss weighing and whether to use discrete time.

## A.4 ADDITIONAL IMPLEMENTATION DETAILS

We use flash attention for all models except as noted below, using the popular Flash-Attention 2 library (Dao, 2023). For all AR models at inference, we use K/V caching and take advantage of specifically optimized functions for this in FlashAttention 2.

In the case of our UniDisc multimodal caching, we instead use Flex Attention (He et al., 2024). This provides the ability to dynamically change the attention mask with minimal performance loss, which is not possible with other flash attention implementations. Benchmarks show that FlexAttention achieves 90% of the performance of FlashAttention-2 (He et al., 2024).

## A.5 LARGE SCALING TRAINING CURVE

We show the training curve for the large scale experiments described in Section 4.6 in Figure 7.

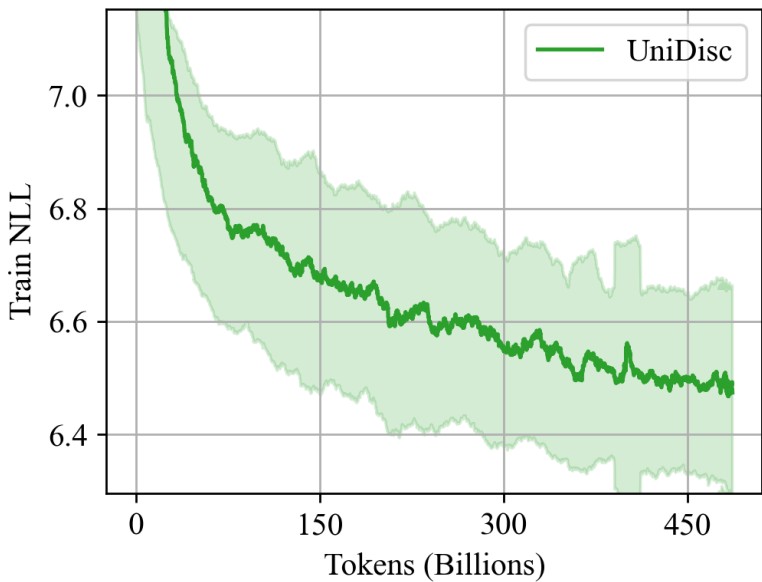

Figure 7: 1.4B Model Training Loss Curve vs Tokens.

## A.6 ADDITIONAL QUALITATIVE RESULTS ON LARGE SCALE MODEL

Here we show more results on tasks such as inpainting, captioning and generation. Note that none of these tasks were explicitly trained or optimized for by our model. This is an intrinsic property due to the nature of UniDisc's unified diffusion based objective.

## A.7 UNDERSTANDING THE EFFECT OF CLASSIFIER FREE GUIDANCE (CFG)

In Table 2, we observe that CFG is a significant factor in the performance difference between UniDisc and the AR baseline. We hypothesize that this is because CFG is most useful in decoding the first few tokens, with diminishing utility in later tokens. To examine this, we look at intermediate predictions by storing $\arg\max p_\theta(x_0 \mid x_t)$ at each sampling step. As an AR model cannot directly capture this distribution without an intractable rollout, we opt to use the same UniDisc model but with an autoregressive inference strategy, decoding from left to right. This allows us to directly compare the performance of different inference strategies and how they interact with classifier-free guidance.

We visualize this in Figure 8, where we visualize the difference between the conditional and unconditional image generated at different %s of decoded tokens. We notice two things: (a) the difference diminishes as more tokens are decoded and (b) UniDisc consistently has higher distances between the logits than AR, which flattens out more quickly.

Intuitively, this means UniDisc extracts much more discriminating signal from CFG compared to AR. We believe this is because UniDisc has much more flexibility to decode tokens initially based on confidence, compared to AR which is forced to decode in a left to right manner and thus, can course correct quickly and more effectively. This can be seen in Table 6, where we selectively apply CFG only on a few steps and notice that CLIP score when CFG is applied on steps 1-3 almost matches applying CFG on all, while applying on the last few steps doesn't affect things much at all.

Given the differences in CFG between UniDisc and AR models, we conduct a hyperparameter sweep over guidance scales in Figure 18. We compute FID and CLIP scores over four datasets, and at both 115/340M parameters. We find that our AR baseline benefits from a weight of $w = 0.5$ but sees far less improvement than UniDisc with CFG. For UniDisc, we choose an overall weight of $w = 1.5$, but note that the CLIP score scales cleanly with the guidance scale, demonstrating the trade-off between visual quality and prompt adherence.

| Steps | CLIP Score |
|-------|------------|
| $[1-3]$ | 0.301 |
| $[12-14]$ | 0.293 |
| $[22-24]$ | 0.283 |
| **All (24)** | 0.312 |

Table 6: Comparing CLIP scores by applying CFG only on specific steps. This shows CFG has the most impact on the initial denoising steps (total steps = 24).

## A.8 GENERATION TIME VS BATCH SIZE

We analyze the quality of the generation versus time in Figure 9. We make a similar observation as in prior work (Ziv et al., 2024; Gat et al., 2024) on discrete diffusion, finding that the ability to obtain predictions with varying sampling steps allows lower latencies. However, with current implementations, KV caching in AR models results in higher throughput as the batch size increases. This tradeoff can be explained by looking at the number of function evaluations (NFEs) and the cost of each in both cases. In AR generation w/KV caching, we have a fixed NFE, but each forward pass is substantially less expensive than in the NAR case. In contrast, in NAR, we can use substantially fewer NFEs, but each is more costly. Modern GPUs only reach peak throughput at larger batch sizes (Chitty-Venkata et al., 2024), or, in other words, as we decrease the batch size, the difference in computation per function evaluation diminishes, resulting in NAR having favorable performance.

## A.9 ANALYZING THE JOINT IMAGE TEXT GENERATION OF UNIDISC

In Figure 16, we visualize how the model iteratively infills both image and text. This raises the question - does UniDisc follow a certain strategy during generation (for example, generating entire background first then moving to subject or generating text first before image), or does it generate everything at once jointly. To analyze this, we take the final model generated image, semantically segment it (using Grounded SAM 2 in our case) and then see which concepts get generated at what timesteps. This is visualized in Figure 17. We find that UniDisc generates all concepts at once

proportional to the overall fraction of the image the concept occupies. We also investigated if the UniDisc has any strong positional bias, such as first generating tokens in the middle and radially filling out. However we find no such positional strategy and that UniDisc is positionally invariant. Intuitively, this means that at any denoising step, all positions are equally likely to be decoded.

### A.10   Zero-shot image editing of UniDisc

A clear benefit of diffusion models is the ability to perform zero-shot editing without specific paired data—which is often difficult to obtain. We demonstrate one such method in Figure 15, showing that UniDisc can automatically improve a user provided image and caption.

We augment real images by overlaying random objects from the COCO dataset. Similarly, we augment captions by asking an LLM to generate purposely incorrect variations. We then randomly mask the image and text inputs and unmask as described above, automatically removing these undesired image artifacts and generating the correct caption. We adopt a best-of-n sampling strategy with n distinct noise masks. We unroll each generation until completion and use the model's own likelihood to select the best generation.

### A.11   Zero-shot length extrapolation of UniDisc

In this section, we demonstrate the ability of UniDisc to perform zero-shot flexible resolution generation thanks to the use of RoPE embeddings on both text and image tokens. UniDisc model was fine-tuned on 512x512 images—resulting in each image using 1024 tokens—but is able to infill at 1024x1024—resulting in 4096 tokens per image—without further training. We demonstrate this in Figure 19.

### A.12   Quantitative Inpainting Comparison w/autoregressive models

To demonstrate the tradeoff between the pre-training objectives of UniDisc and AR models, we evaluate both models on inpainting. We fine-tune the 340M parameter AR model on a standard set of multimodal datasets (CC12M, Recap-DataComp-1B, LAION 400M) and evaluate UniDisc in a zero shot manner—without any fine-tuning. Specifically, for the AR model, we use a linear masking schedule for the prefix sequence consisting of a randomly masked text and image pair and then predict and supervise the clean sequence, doubling the overall sequence length. In Figure 20, we evaluate at multiple noise levels, showing the degradation in performance as the original sequence is increasingly masked.

### A.13   Comparison with recent mulitmodal models

In Table 7, we evaluate UniDisc on the popular GenEval Ghosh et al. (2023) benchmark which looks at how well a generated image adheres to the prompt in terms of a set of predefined attributes (e.g., color, positioning). In Table 8, we compare FID on the popular MS-COCO 30K (Chen et al., 2015) dataset on MJHQ-30K (Li et al., 2024a), which contains a higher propertion of highly-aesthetic images.

| Method | Sing. Obj. | Two Obj. | Counting | Colors | Position | Color Attr. | Overall |
|---|---|---|---|---|---|---|---|
| SDv1.5 (Rombach et al., 2022) | 0.97 | 0.38 | 0.35 | 0.76 | 0.04 | 0.06 | 0.43 |
| CoDI (Tang et al., 2024) | 0.89 | 0.16 | 0.16 | 0.65 | 0.02 | 0.01 | 0.31 |
| Lumina-mGPT (Liu et al., 2024) | - | - | - | - | - | - | 0.32 |
| UniDisc | 0.92 | 0.47 | 0.15 | 0.67 | 0.13 | 0.19 | 0.42 |

Table 7: We evaluate UniDisc on the GenEval (Ghosh et al., 2023) benchmark.

| Method | MSCOCO-30K FID $\downarrow$ | MJHQ-30K FID $\downarrow$ |
|---|---|---|
| SDv1.5 (Rombach et al., 2022) | 11.12 | - |
| CoDi (Tang et al., 2024) | 22.26 | 19.87 |
| UniDisc (Ours) | 23.86 | 18.67 |

Table 8: We evaluate the 1.4B version of UniDisc on FID. We use evaluate on MS-COCO 30K (Chen et al., 2015) and MJHQ-30K (Li et al., 2024a).

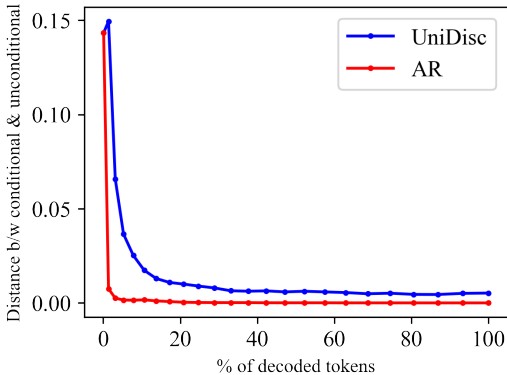

Figure 8: L2 distance between unconditional and conditional logits on currently masked tokens as sampling steps increase.

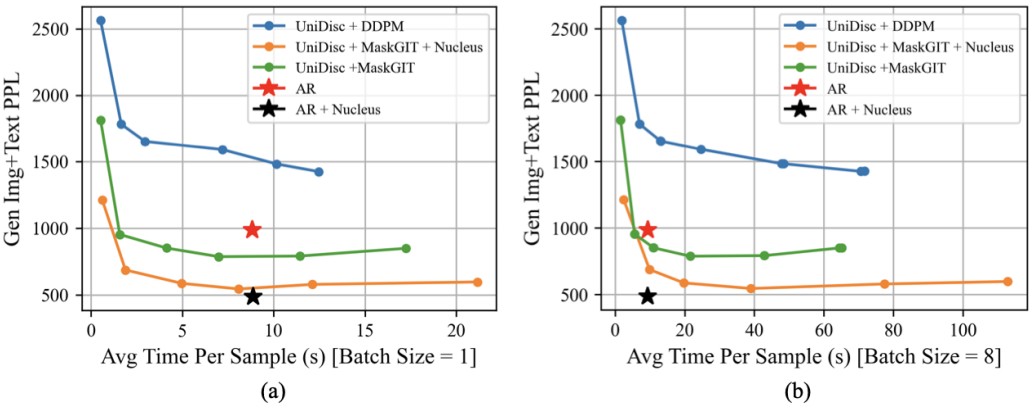

Figure 9: Generative Perplexity vs. Time with various models and sampling strategies.

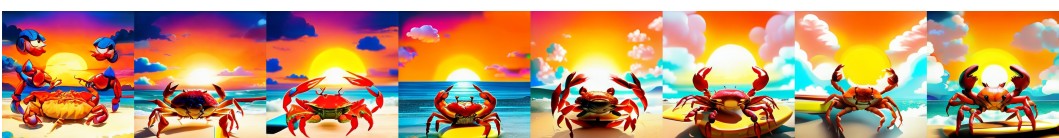

Figure 10: We show the effect of classifier-free guidance from left-to-right, starting with $w = 0$, and increasing linearly to $w = 8$ on the right, where output logits are $l_{\text{cfg}} = (1 + w)l_{\text{cond}} + w * l_{\text{uncond}}$. Caption: "crab meditating, surfboard, orange sun setting, rainbow clouds, zen beach"

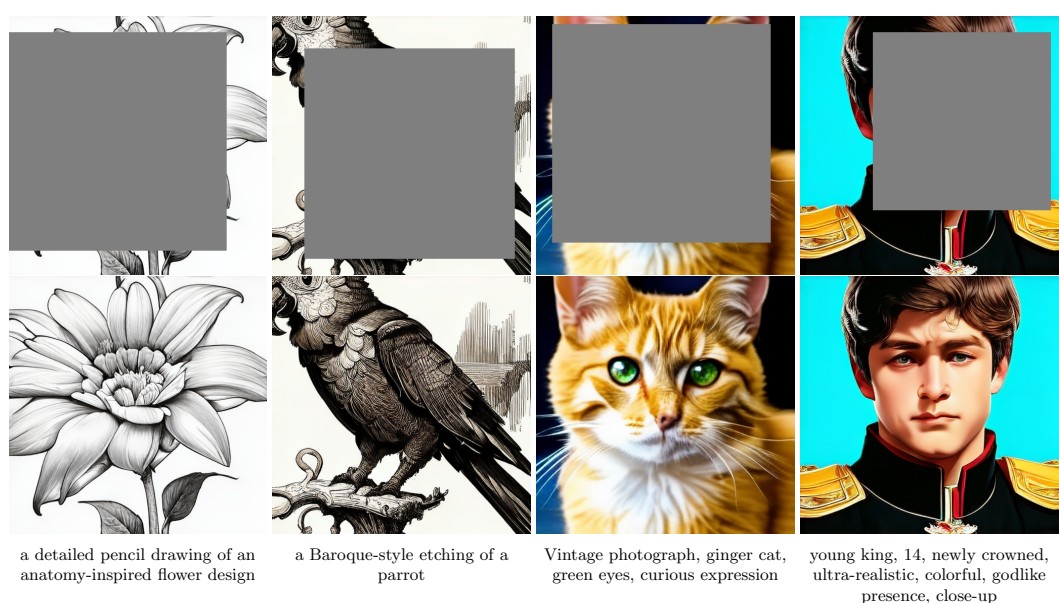

Figure 11: Zero-shot text-conditioned inpainting. UniDisc inpaints a masked region given a user-provided text prompt.

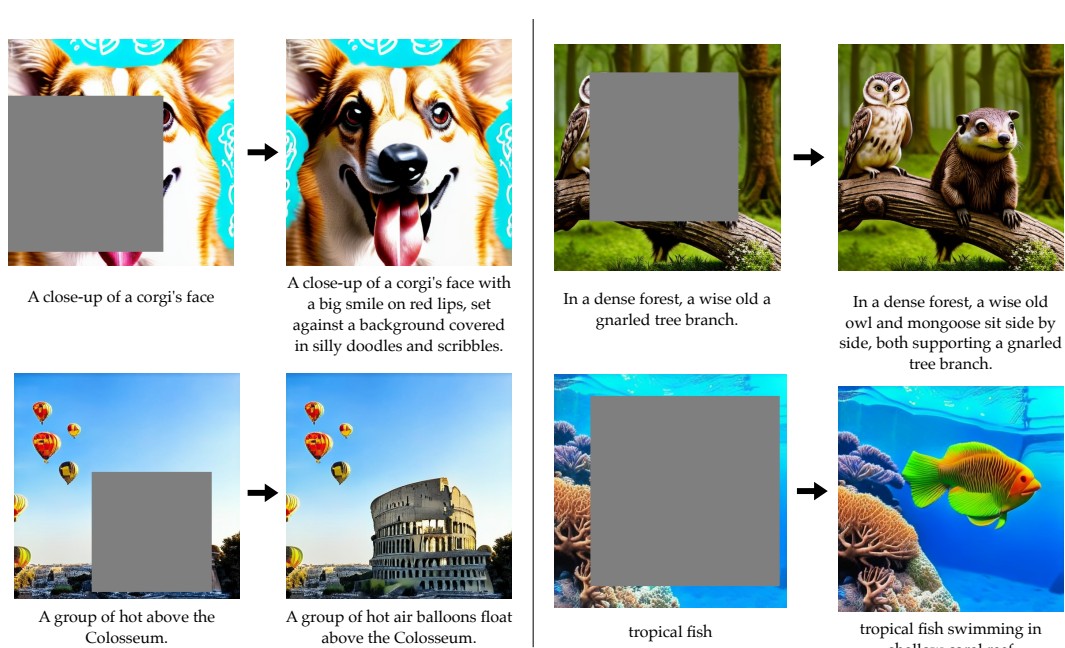

Figure 12: Zero-shot multimodal inpainting. UniDisc jointly inpaints in both image and text spaces.

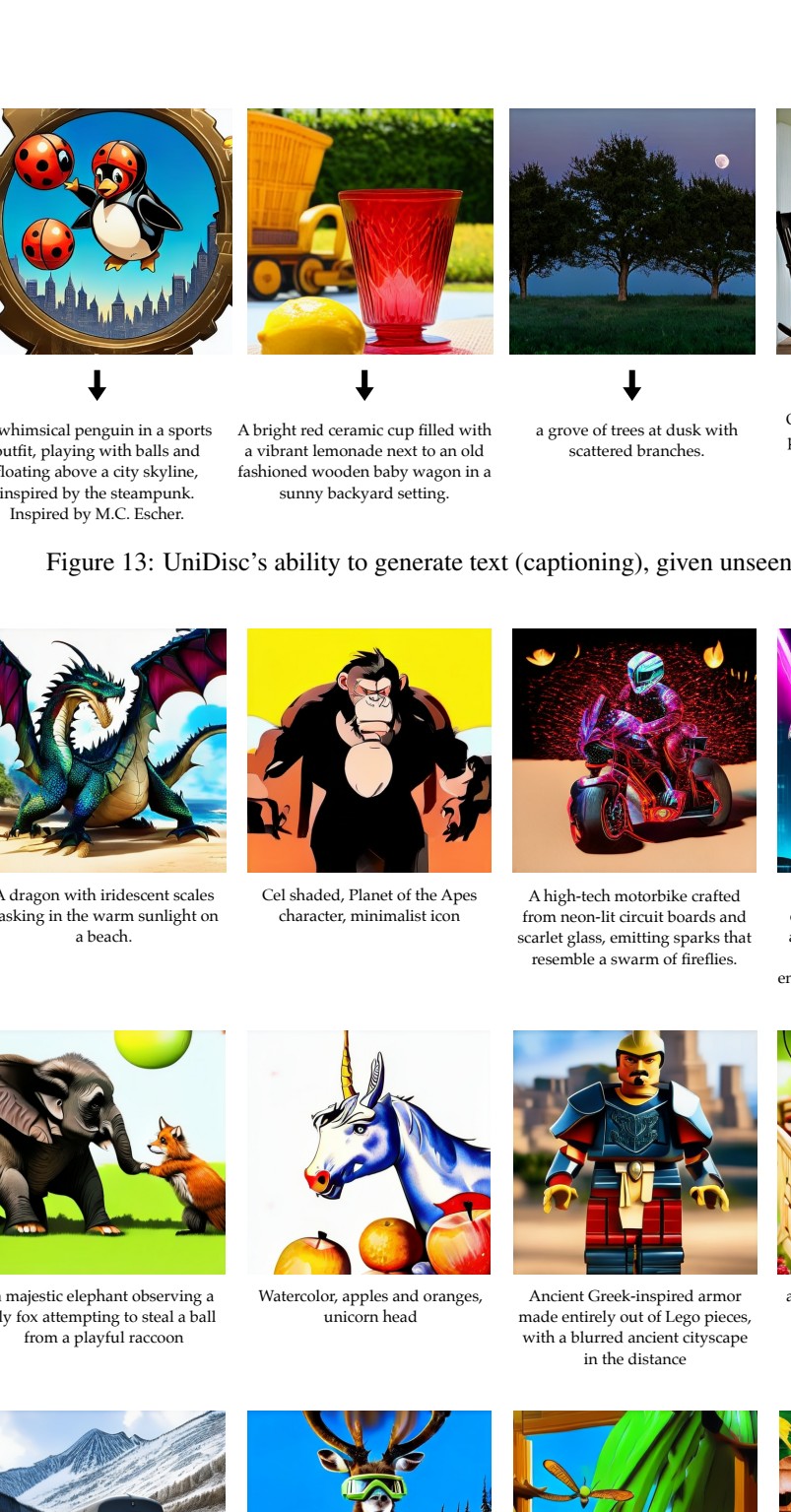

Figure 13: UniDisc's ability to generate text (captioning), given unseen image as input.

Figure 14: UniDisc's ability to generate an image, given unseen text as input.

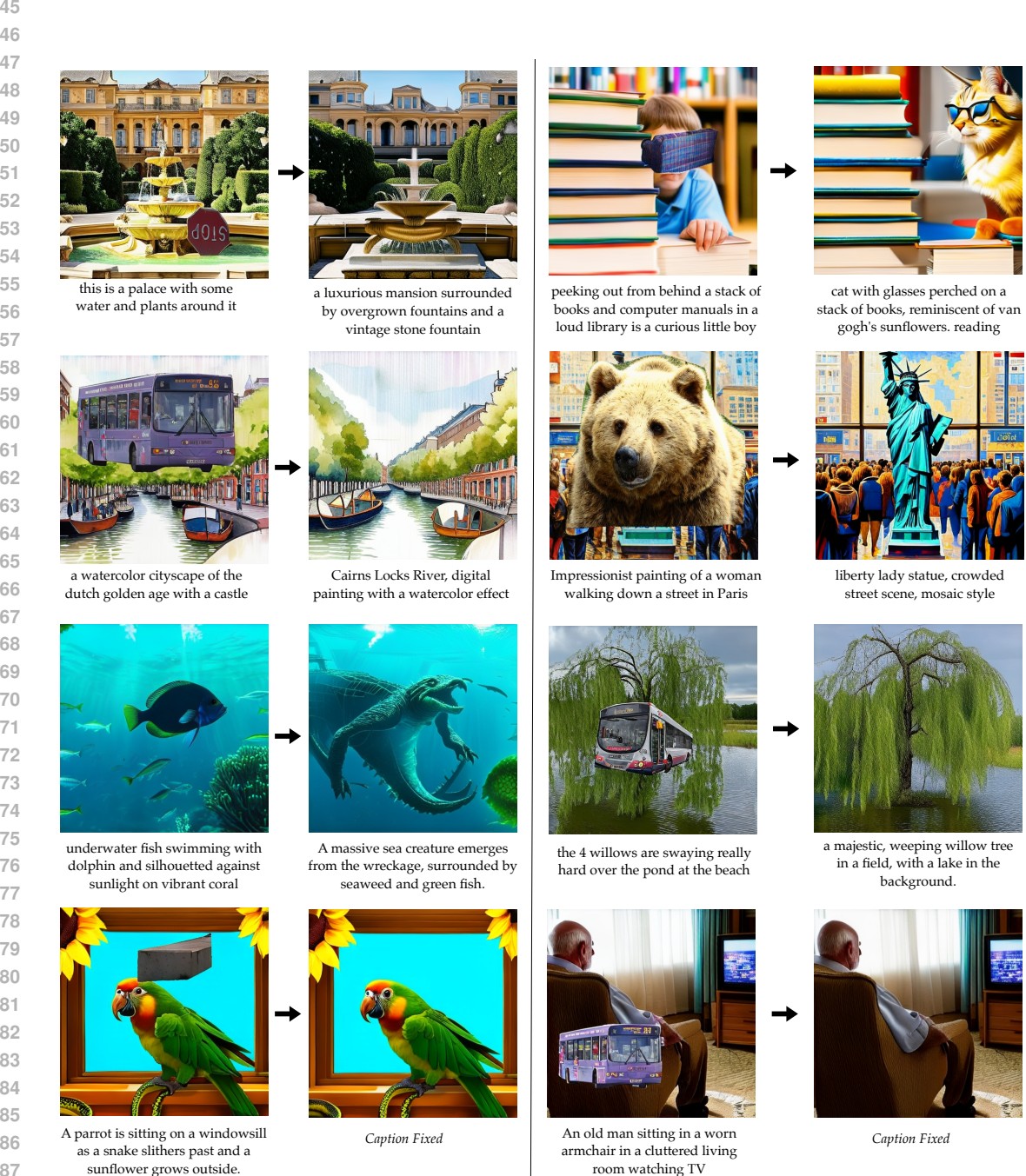

Figure 15: Zero-shot multimodal editing. We provide a *clean* image and text pair and UniDisc automatically unhances both the image and text. In the final row, we fix the text and allow only the image to change.

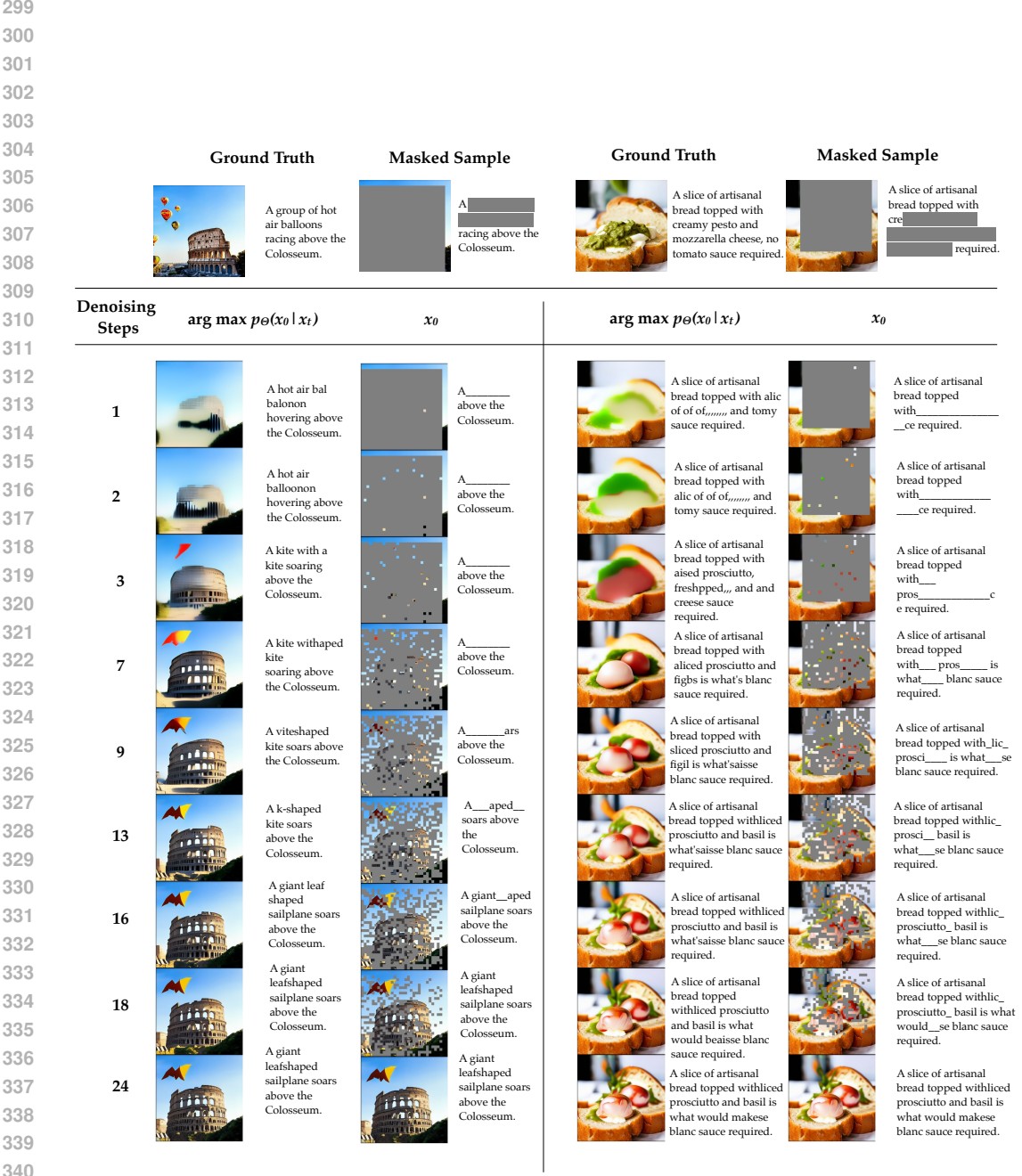

Figure 16: We show how UniDisc jointly infills both image and text. $\arg\max p_\theta(x_0 \mid x_t)$

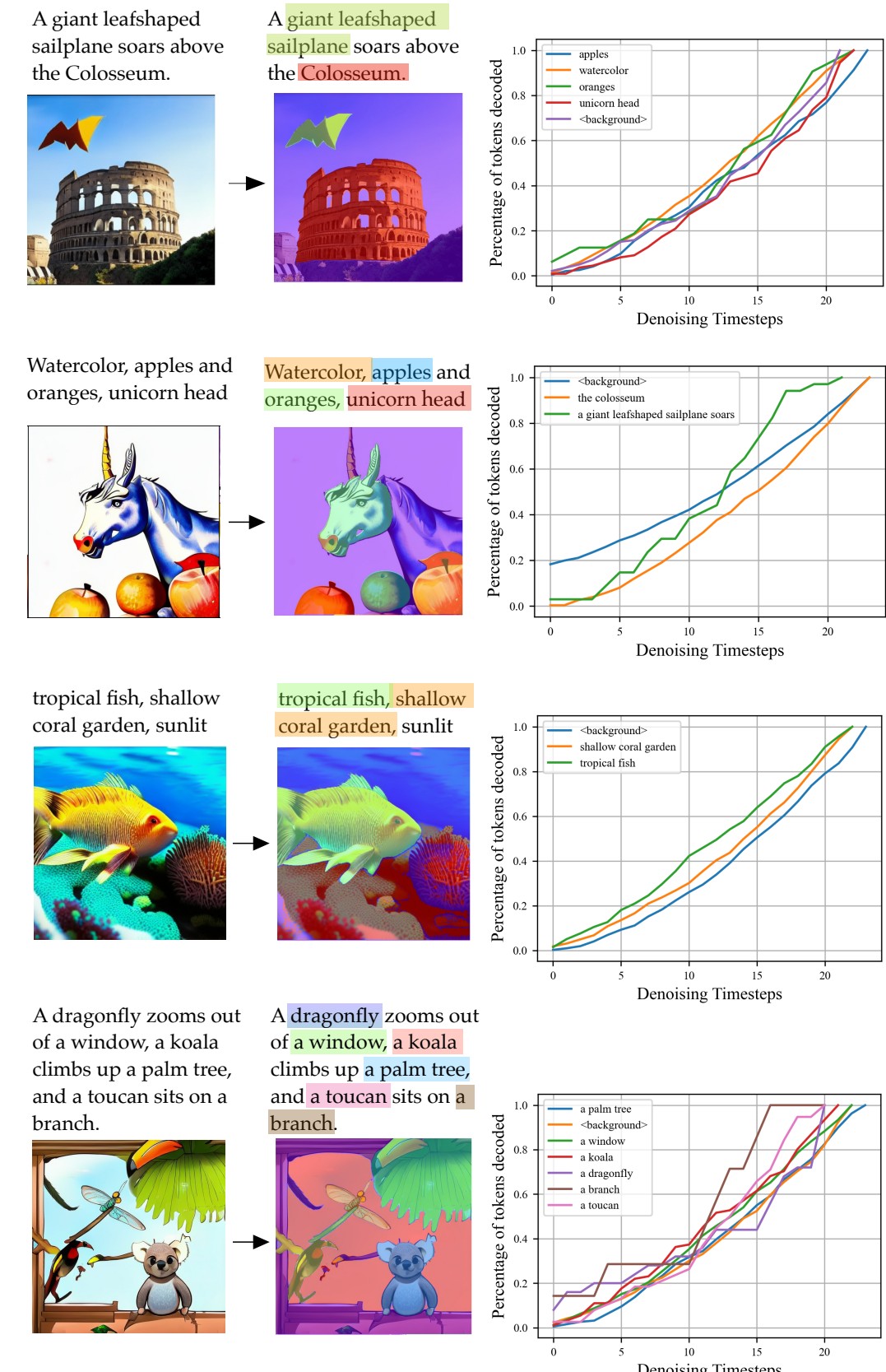

Figure 17: We show how UniDisc uniformly generates all concepts at once.

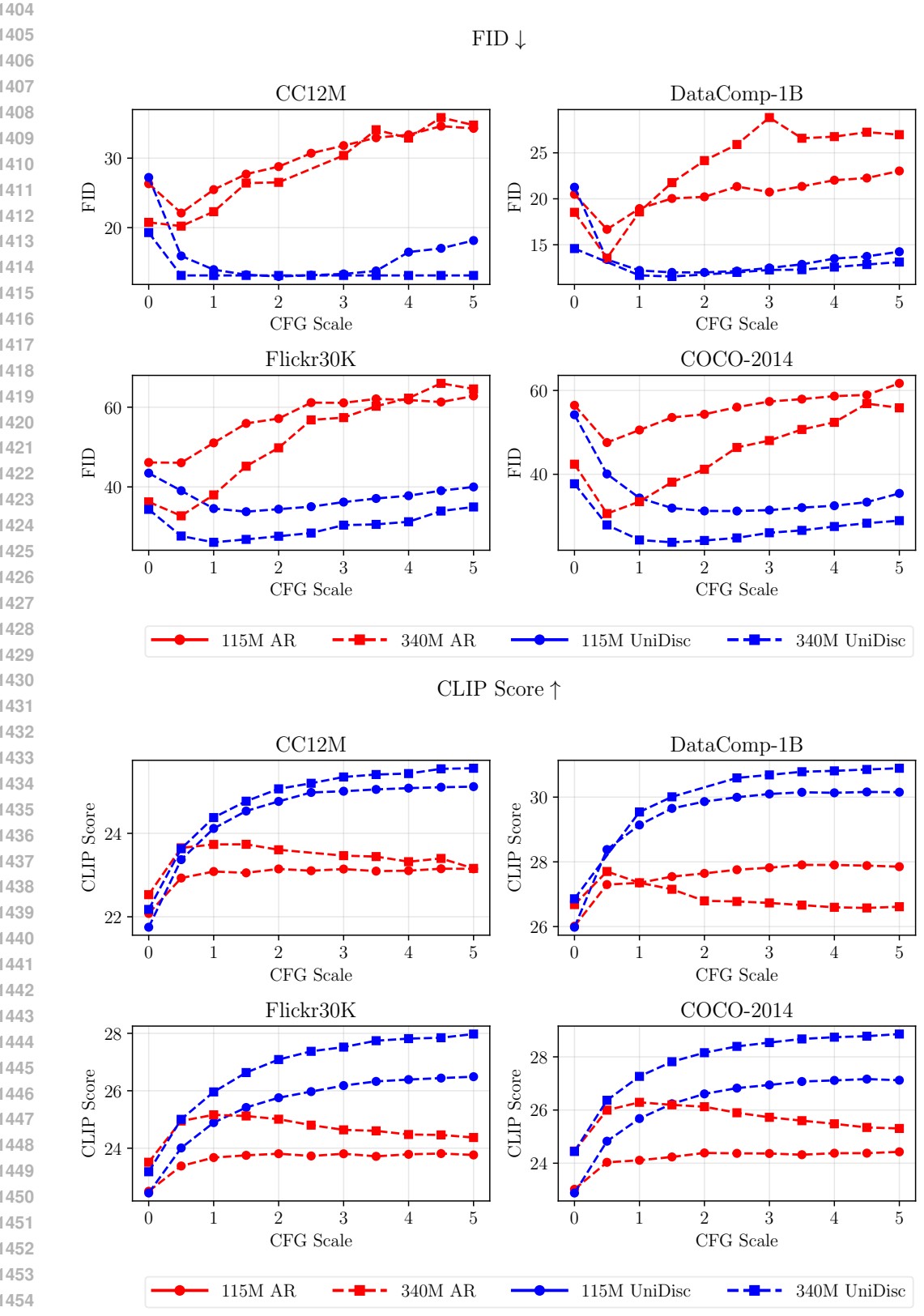

Figure 18: We ablate the CFG weight for both model scales on both FID and CLIP metrics. We find that AR is more sensitive to the CFG weighting, with a narrower optimal range.

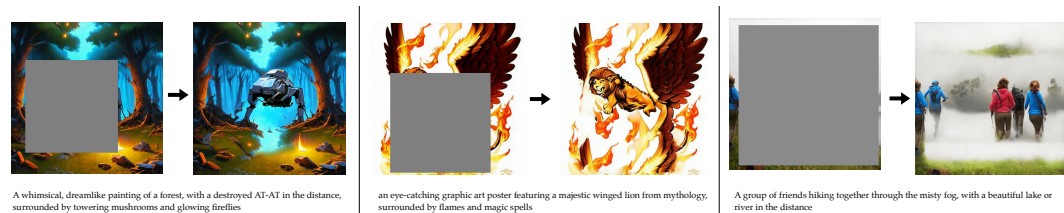

Figure 19: We train UniDisc on 512x512 resolution images but demonstrate zero-shot inpainting at 1024x1024.

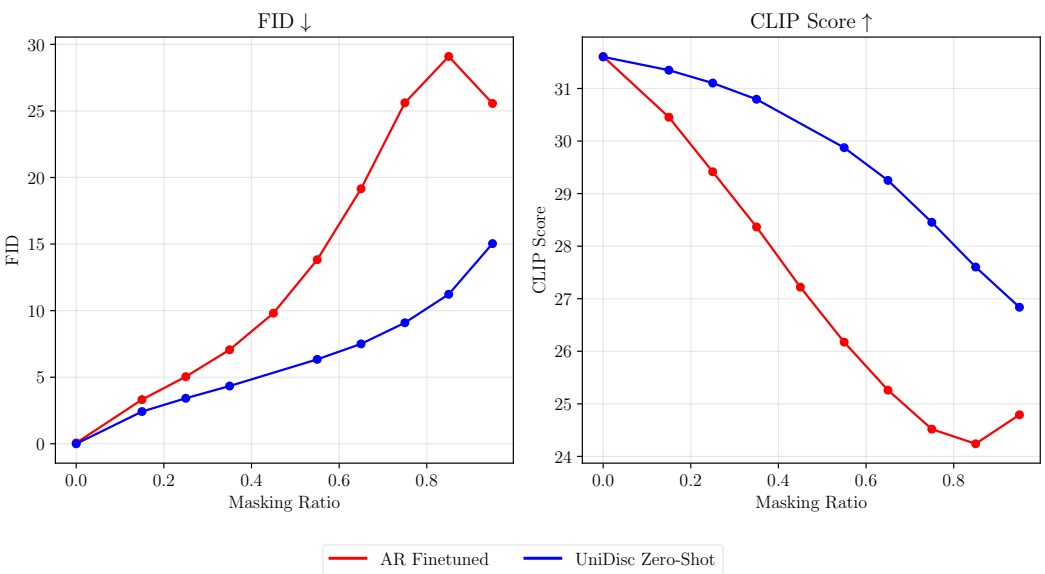

Figure 20: We compare UniDisc with an AR model fine-tuned for joint inpainting and evaluate on a subset of DataComp1B.

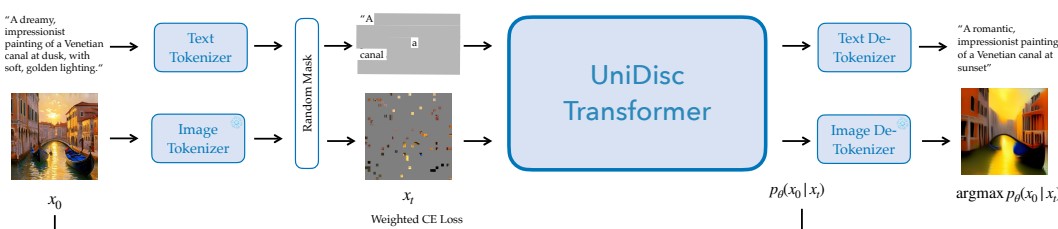

Figure 21: UniDisc is a unified multimodal discrete diffusion model that can jointly process and generate text and images. First, each modality is converted into a sequence of discrete tokens and we randomly replace a subset of these tokens with the [MASK] token according to a noise schedule and denoted in the figure with grey boxes. We jointly denoise the image and text and supervise with a weighted cross-entropy loss. At inference time we begin with a set of [MASK] tokens and iteratively unmask tokens. We visualize the intermediate generations during inference in Figure 16.

