# OpenReview forum: "Unified Multimodal Discrete Diffusion"
_ICLR.cc/2025/Conference — Submitted to ICLR 2025_

### Official Review · Reviewer_JxTt · 2024-10-27

**Soundness:** 3
**Presentation:** 4
**Contribution:** 3
**Rating:** 6
**Confidence:** 4

**Summary:**

The paper presents Unified Multimodal Discrete Diffusion (UniDisc), a model that leverages discrete diffusion for joint text-image generation tasks. UniDisc uses discrete diffusion by masking tokens and iteratively denoising them， which allows for enhanced controllability, quality-diversity trade-offs, and efficient inpainting across modalities. UniDisc is designed to overcome AR limitations, such as inefficiency in image generation and lack of flexibility in conditional generation. The model introduces a modality-specific caching mechanism to address varying token sampling needs for images and text, resulting in improved inference efficiency. Evaluation across multiple datasets shows that UniDisc outperforms AR baselines in generation quality, inference efficiency, and retrieval tasks, demonstrating its potential for scalable multimodal applications.

**Strengths:**

* Overall I find that the writing is clear and well-structured, making it easy for readers to follow the arguments and understand the key points.
* The exploration of discrete diffusion for multimodal tasks is valuable. The proposed framework demonstrates the potential of various multimodal tasks, such as text-image jointing inpainting and retrieval.
* The paper shows sufficient ablation studies to verify the effectiveness of each design choice of UniDisc.

**Weaknesses:**

* This paper mainly applies discrete diffusion to multimodal tasks and integrates some existing techniques from previous papers, such as classifier-free guidance and sampling schedules from MaskGiT, D3PM, etc. Therefore, the technical contribution of this paper is limited.
* However, if this paper is considered an empirical paper, it lacks a comprehensive comparison with other advanced methods on important benchmarks, such as the original Chameleon, EMU, Next-GPT, and CoDI. Additionally, it is well known that the greatest advantage of AR models lies in scaling laws, while many studies have found that scaling for Non-AR models (Discrete Diffusion) is a bottleneck. Therefore, training only a 100M model and comparing it with a single AR baseline is not very convincing. More fair comparison experiments should be conducted across different model sizes and data scales.

**Questions:**

* In Table 2, the FID significantly drops after applying CFG to the AR baseline, while UniDisc gains a significant performance boost. This is an unusual result that CFG greatly hurts the FID of AR models. Could you please explain this observation and provide more details on the implementation of the AR model?
* In Figure 4, the paper compares the inference efficiency between UniDisc and AR baseline. Does the AR baseline integrate techniques like Flash-Attention, which can greatly boost the inference speed?

---

> ### Author Response · Authors · 2024-11-21
> **Official Comment by Authors (1/3)**
>
> Thank you for your constructive feedback and helpful suggestions. We address your concerns point-by-point below.
>
> **Q4.1: Limited Technical Contribution**:
>
> Our technical contributions are as follows:
>
> i) We present the first unified discrete diffusion model that can jointly inpaint and edit in image-text space.
>
> ii) We present a multimodal caching strategy for discrete diffusion models that increases the model’s throughput by a significant margin.
>
> iii) Prior methods in discrete diffusion focus only on unimodal tasks and find very different tradeoffs. For instance, MaskGIT focuses on images and uses cosine schedule, confidence based sampling, discrete time steps and standard masked prediction loss. On the other hand D3PM focuses on text and uses a linear schedule, random sampling, continuous timesteps and an ELBO loss. In this work we abate all of these design choices and find the best working combinations for unified multimodal modeling for discrete diffusion.
>
> iii) Lastly, we are the first work to densely evaluate unified discrete diffusion against autoregressive models, across multiple model sizes, dataset sizes and metrics.
>
> \
> **Q4.2: Comparisons over different model sizes and data scales:**
>
> Thank you for your suggestion. To satisfy your concern,  we have run experiments with our model and AR baseline having 340M parameters each. In addition, we also trained on a different data scale, where we increased the dataset from 11B to 60B unique tokens. We did this by including additional image-text pairs from the Recap-DataComp-1B and LAION 400M datasets. **The results have been updated in Table 2 in the main paper and are shown below.** We find similar results in both models after scaling, with particular improvement in out-of-domain datasets.
>
> |  | CC12M | DataComp | Flickr | COCO |
> | :---- | :---- | :---- | :---- | :---- |
> | **Text to Image \- FID** |  |  |  |  |
> | Chameleon 115M w/o CFG | 26.32 | 20.49 | 46.13 | 56.46 |
> | Chameleon 340M w/o CFG | 20.75 | 18.53 | 36.24 | 42.41 |
> | Chameleon 115M w/ CFG (0.5) | 22.10 | 16.68 | 46.06 | 47.58 |
> | Chameleon 340M w/ CFG (0.5) | 20.22 | 13.55 | 32.74 | 30.62 |
> | UniDisc 115M w/o CFG | 27.22 | 21.26 | 43.46 | 54.21 |
> | UniDisc 340M w/o CFG | 19.28 | 14.59 | 34.37 | 37.73 |
> | UniDisc 115M w/ CFG (1.5) | **13.21** | **12.00** | **33.79** | **31.94** |
> | UniDisc 340M w/ CFG (1.5) | **13.11** | **11.55** | **26.83** | **23.77** |
> | **Image to Text \- CLIP** |  |  |  |  |
> | Chameleon 115M w/o CFG | 22.08 | 26.01 | 22.50 | 23.02 |
> | Chameleon 340M w/o CFG | 22.53 | 26.68 | 23.51 | 24.46 |
> | Chameleon 115M w/ CFG (0.5) | 22.93 | 27.30 | 23.38 | 24.03 |
> | Chameleon 340M w/ CFG (0.5) | 23.65 | 27.70 | 24.95 | 25.99 |
> | UniDisc 115M w/o CFG | 21.75 | 25.98 | 22.44 | 22.88 |
> | UniDisc 340M w/o CFG | 22.18 | 26.86 | 23.18 | 24.44 |
> | UniDisc 115M w/ CFG (1.5) | **24.54** | **29.65** | **25.42** | **26.24** |
> | UniDisc 340M w/ CFG (1.5) | **24.77** | **30.01** | **26.63** | **27.82** |

---

> ### Author Response · Authors · 2024-11-21
> **Official Comment by Authors (2/3)**
>
> **Q4.3: Suggested Baselines**
>
> We appreciate the suggestions for baselines. These methods are indeed impressive, but we do not have the compute resources in order to re-train these from scratch for a fair comparison. For example, the smallest of the listed models, CoDI$^1$ consists of 5.3B parameters and uses pretrained StableDiffusion \+ ViT \+ GPT2 weights.
>
> We plan to compare our 1.4B model to CoDI and separately train a continuous diffusion baseline at a smaller scale for a fair comparison. Although this differs from the training stages of CoDI, we believe this will be a valuable baseline.
>
> *Emu3 \[1\]*: This work largely follows Chameleon \[4\] — although exact data mixtures, tokenization hyperparameters, and training stages vary, the overall objective and general architecture remains the same. We note that the focus of their work was on high-quality data—which is not released—and on post-training, both of which would equally benefit AR and NAR models.
>
> *NExT-GPT \[3\]*: This work attempts to "leverag\[e\] ... existing well-trained high-performing encoders and decoders" and does so by having an LLM generate conditioning embeddings for an image decoder, freezing all weights except small projection and adaptation layers. We do not compare to NExT-GPT as their model comprises 9.6B parameters, making a comparison to our models unfair.
>
> If we were to train from scratch at a feasible scale, there are numerous unclear choices for pre-training—e.g., allocation of parameters and training compute between the LLM and image decoder.
>
> *CoDI \[2\]*: This work attempts to enable multimodal generation for unseen pairings by aligning a set of encoders with frozen modality decoders. Notably, these include a text diffusion U-Net that conditions a pre-trained GPT-2 decoder and a fine-tuned Stable Diffusion image decoder.
>
> We note that the above methods are intended to address: (i) how to integrate task-specific models and (ii) the performance of these combined models given varying data pairings and mixtures.
>
> **Q4.4: Why CFG hurts FID for the AR model?:**
>
> The CFG results in the paper were obtained with the CFG scale set to 4 for all models, following the optimal value in CM3Leon \[5\]. Since submission, we have run a sweep across CFG scales across all datasets and model sizes, evaluating FID and CLIP scores. We found optimal guidance scale values for AR and UniDisc, which improve performance for both models—but the performance of UniDisc remains higher than the AR model, as you can see at [unidisc-diffusion.github.io/#cfg_ablation](https://unidisc-diffusion.github.io/#cfg_ablation) and in Figure 18 of the updated manuscript.
>
> Since submission, we conducted an analysis to study why CFG doesn’t significantly improve Chameleon’s performance as it does for UniDisc (Section A.7 in the updated manuscript). CFG enforces the model to go in the direction of the conditional predictions while going in the opposite direction of its unconditional predictions. We find that in autoregressive models, the unconditional predictions start becoming very similar to its conditional predictions after the first token gets decoded. This is because the next token to be predicted is highly correlated to its previous token due to the left-to-right decoding strategy. The high correlation makes it simpler for the unconditional model to predict the next token, thus making the guidance play a smaller role in later steps. In UniDisc this doesn’t happen, as the model predicts tokens in random locations in the sequence, making it tougher for the unconditional predictions to match the conditional predictions. We visualize this trend [here](https://unidisc-diffusion.github.io/#cfg_unidisc_ar_analysis) (Figure 8 of the updated manuscript) by plotting the distance between the unconditional and conditional tokens as more tokens are decoded.
>
> \----------------------------------------------------------------------------
> 1: We only count the parameters used for text and image encoding/decoding.
>
> \[1\] X. Wang et al., “Emu3: Next-Token Prediction is All You Need,” Sep. 27, 2024, arXiv: arXiv:2409.18869.
>
> \[2\] Z. Tang, Z. Yang, C. Zhu, M. Zeng, and M. Bansal, “Any-to-Any Generation via Composable Diffusion,” May 19, 2023, arXiv: arXiv:2305.11846.
>
> \[3\] S. Wu, H. Fei, L. Qu, W. Ji, and T.-S. Chua, “NExT-GPT: Any-to-Any Multimodal LLM,” Jun. 25, 2024, arXiv: arXiv:2309.05519.
>
> \[4\]: Chameleon Team, “Chameleon: Mixed-Modal Early-Fusion Foundation Models,” May 16, 2024, arXiv: arXiv:2405.09818.
>
> \[5\] L. Yu et al., “Scaling Autoregressive Multi-Modal Models: Pretraining and Instruction Tuning,” Sep. 05, 2023, arXiv: arXiv:2309.02591.

---

> ### Author Response · Authors · 2024-11-21
> **Official Comment by Authors (3/3)**
>
> **Q4.5: Does AR integrate flash attention?**
>
> We use flash attention for all models, specifically we use the popular Flash-Attention 2 library \[1\]. For all AR models at inference, we use K/V caching and take advantage of specifically optimized functions for this in FlashAttention 2\.
>
> In the case of our UniDisc multimodal caching, we instead use Flex Attention \[2\]. This provides the ability to dynamically change the attention mask with minimal performance loss, which is not possible with other flash attention implementations. Benchmarks show that FlexAttention achieves \~90% of the performance of FlashAttention-2 \[2\].
>
> We have added this description to Sec A.4 of the supplementary materials.
>
> \----------------------------------------------------------------------------
> \[1\] T. Dao, “FlashAttention-2: Faster Attention with Better Parallelism and Work Partitioning,” Jul. 17, 2023, arXiv: arXiv:2307.08691
>
> \[2\]: H. He, D. Guessous, Y. Liang, and J. Dong, “FlexAttention: The Flexibility of PyTorch with the Performance of FlashAttention,” PyTorch.

---

> ### Comment · Reviewer_JxTt · 2024-11-21
> **Reply to Author Comment**
>
> Thank you for your response. I believe this is an excellent rebuttal and could serve as a template for others. The rebuttal effectively addressed nearly all of my questions, particularly the analysis of the impact of CFG, which provided some interesting insights. However, I still believe the paper has a few significant issues:
>
> * While the authors highlight several "firsts" as contributions of this paper, the majority of the components are based on existing techniques. For example, the proposed multimodal caching strategy for discrete diffusion models essentially applies two different schedules to two distinct modalities. And I notice that the UniD3 [1] looks very similar to this paper. Could you elaborate the differences with this paper?
>
> * A major concern remains that the generative performance of the model falls significantly short of the state-of-the-art across various tasks. The quality of the generated images exhibits very noticeable "MaskGiT-style" artifacts. The paper does not compare against any unified multimodal generative models or specialist models for image or text generation, providing only an autoregressive model as a baseline. Additionally, the model size of 300M parameters is smaller than many models trained on ImageNet, let alone text-to-image or multimodal generative models. From my own recent experience training MaskGiT/Discrete Diffusion models ranging from 100M to 3B parameters, I’ve observed that scaling effects tend to saturate beyond 1B parameters. Therefore, I would have liked to see a more in-depth exploration of scaling for discrete diffusion models.
>
> That said, I fully understand that these expectations may be beyond the scope of a paper that is the first to explore multimodal discrete diffusion, especially given the computational resources required. Furthermore, I greatly appreciate the interesting applications presented in the rebuttal, which showcase the potential of multimodal discrete diffusion as a promising unified architecture that could compete with diffusion, autoregressive, and Transfusion approaches.
>
> As a result, I am inclined to raise my score.
>
> [1] Hu, Minghui, et al. "Unified discrete diffusion for simultaneous vision-language generation." arXiv preprint arXiv:2211.14842 (2022).

---

> ### Author Response · Authors · 2024-11-27
>
> We greatly appreciate your comments and are glad that you found our additional experiments beneficial!
>
> \
> **Q4.6**: Multimodal Caching
> > While the authors highlight several "firsts" as contributions of this paper, the majority of the components are based on existing techniques. For example, the proposed multimodal caching strategy for discrete diffusion models essentially applies two different schedules to two distinct modalities.
>
> We are the first to propose a multimodal caching strategy in discrete diffusion. This strategy reduces the overall inference compute by allowing each modality to use a variable amount of computation. UniD3 applies the same schedule across text and image modality and doesn’t do any caching—all modalities use a linear schedule.
>
> As we describe in the submission, each modality benefits from a different number of optimal generation steps. To account for this, we propose our multimodal caching where the model learns both conditional and unconditional embeddings, the latter of which can be cached to allow for variable frequencies between modalities. This does not affect the architecture and is implemented through random dropout on the model’s attention mask.
>
> \
> **Q4.7**: Relevant Work
> > *And I notice that the UniD3 \[1\] looks very similar to this paper. Could you elaborate the differences with this paper?*
>
> As we describe on Lines 123-127, UniD3 is indeed relevant to our paper. However there are major differences between both the works:
>
> (1) They use a hybrid of a uniform and absorbing schedule, while we only use the absorbing schedule. We find the absorbing schedule to be a lot more training efficient than the uniform schedule. For instance we find that our 115M parameter model has an overall NLL of 6.1 at 60k steps with an absorbing schedule but 6.59 with a uniform schedule—with similar findings by several previous works including \[1\].
>
> (2) They decouple the modalities within the model, with a “mutual-attention” mechanism between modalities, and in loss computation. Instead, we simply use standard self-attention with a single weighted cross-entropy loss term (Eq 3\) and show that this scales to a 1.4B parameter model.
>
> Further we couldn’t compare against their model, as we were not able to reproduce their reported results using their available code \[3\]. The full training code was not made public, with missing hyperparameters such as the ratio of fully unmasking one modality and numerical precision issues with the use of uniform diffusion over the vocab size. We tried our best to fix these bugs but we were to successfully reproduce in a way consistent with their original design.
>
> Aside from this, our work includes comparison to AR models on multiple tasks, ablation of strategies in both image and text diffusion literature, multimodal caching, and demonstration of automatic editing, which we believe are useful technical contributions to the field of unified generative modeling.
>
> \
> **Q4.8**: Baseline Comparisons
> > The paper does not compare against any unified multimodal generative models or specialist models for image or text generation, providing only an autoregressive model as a baseline.
>
> Thank you for the suggestion. We have added evaluations of our model on GenEval, MSCOCO and MJHQ in our updated submission to provide a better understanding of the model’s capabilities.
>
> \----------------------------------------------------------------------------
> \[1\] J. Austin, D. D. Johnson, J. Ho, D. Tarlow, and R. van den Berg, “Structured Denoising Diffusion Models in Discrete State-Spaces,” 2023, arXiv:2107.03006
>
> \[2\] A. Lou, C. Meng, and S. Ermon, “Discrete Diffusion Modeling by Estimating the Ratios of the Data Distribution,” 2024, arXiv:2310.16834
>
> \[3\] [UniD3 Code](https://github.com/mhh0318/UniD3)

---

> > ### Comment · Reviewer_JxTt · 2024-11-27
> >
> > Thanks for this further response. Most of my concerns are addressed. Although the current version lags behind sota continuous diffusion models, I believe its promising capabilities as a multimodal foundational framework.

---

### Official Review · Reviewer_8LSP · 2024-10-31

**Soundness:** 3
**Presentation:** 3
**Contribution:** 3
**Rating:** 5
**Confidence:** 3

**Summary:**

The paper introduces UniDisc, a unified multimodal discrete diffusion model that operates on both text and image modalities. Unlike traditional autoregressive (AR) models, UniDisc leverages discrete diffusion, which masks tokens and then learns to reconstruct them. This approach offers several advantages, including enhanced controllability, joint image-text inpainting, and efficiency in inference.

**Strengths:**

1. This paper introduces a unified multimodal discrete diffusion approach designed to address the limitations of both continuous diffusion and autoregressive (AR) models. Unlike continuous diffusion models, which struggle with discrete data like text, UniDisc leverages discrete noise through token masking, enabling it to seamlessly handle both images and text in a single model. By unifying the generation process for different modalities, UniDisc allows for efficient, high-quality multimodal outputs with better control over trade-offs between quality and diversity.

2. The authors conduct extensive experiments across various tasks, datasets, and evaluation metrics to assess UniDisc's capabilities. These experiments include unconditional and conditional generation tasks, which highlight UniDisc's ability to generate both text and images independently and conditioned on each other. Through these experiments, UniDisc demonstrates superior performance over AR baselines, especially when using classifier-free guidance (CFG) in conditional generation tasks, which significantly boosts its quality over AR models.

**Weaknesses:**

1. It is recommended to add a visual diagram showing the step-by-step workflow of your proposed method. A clear pipeline figure would make your methodology much easier to follow.

2. While the authors include some visualization results in the appendix, the paper would benefit from more comprehensive examples of the model's outputs. Since this is a generation-focused paper, it is suggested to include a diverse set of generated images and texts directly in the main manuscript. Showing results across different scenarios would help readers better evaluate the method's capabilities and performance.

**Questions:**

Please refer to the weaknesses section.

---

> ### Author Response · Authors · 2024-11-21
>
> Thank you for your constructive feedback and valuable suggestions. We address your concerns point-by-point below.
>
> **Q3.1: Model Diagram**: We appreciate the suggestion to add a workflow figure. Please find the diagram here: https://unidisc-diffusion.github.io/#overall_architecture
>
> **Q3.2: Additional Visualizations**:
>
> We have added numerous additional visualizations to our website at [unidisc-diffusion.github.io](unidisc-diffusion.github.io) which includes text-to-image, image-to-text and joint inpainting. We have also added these to the appendix in manuscript.
>
> In addition, we demonstrate UniDisc’s ability to jointly edit an image-text pair using its likelihood as a scoring function, with visualizations shown at [unidisc-diffusion.github.io/#zero_shot_editing](https://unidisc-diffusion.github.io/#zero_shot_editing) and in Section A.10 of the updated manuscript). To the best of our knowledge, this capability hasn't been shown previously in any other generative model.
>
> We perform further analysis of UniDisc at [unidisc-diffusion.github.io/#generation_analysis](https://unidisc-diffusion.github.io/#generation_analysis) and in Section A.9 of the updated manuscript, visualizing the generation of UniDisc at intermediate denoising steps, made possible by the prediction of $x\_0$ at each step. We observe that the model generates uniformly over concepts and modalities. In AR models this cannot happen as the model generates in a fixed order (e.g., text first, then raster-order), and thus the model cannot jointly reason over modalities and multiple parts of the image.

---

> ### Author Response · Authors · 2024-11-25
>
> Dear reviewer, we look forward to your response to our rebuttal. The discussion period is nearing its end, we'd appreciate it if you could review it and share your feedback before the discussion period closes. Thank you.
>
> We plan to update the submission shortly with a step-by-step diagram. However, we have added many comprehensive visualizations, in the updated submission and on the website, that are relevant to your comments.

---

> ### Author Response · Authors · 2024-11-27
>
> Dear Reviewer,
>
> Just gently checking once again to see if you had an opportunity to look at our rebuttal. We have updated our submission with a step-by-step diagram (Figure 21) and numerous additional visualizations, also available on our website: [unidisc-diffusion.github.io](https://unidisc-diffusion.github.io/)
>
> We hope to have addressed all your concerns but if you have any more questions that prevent you from accepting the paper, please let us know as the deadline for experiments is near.

---

> ### Author Response · Authors · 2024-12-01
>
> Dear reviewer, we are once again gently checking to see if you had an opportunity to look at our rebuttal. We hope to have addressed all your concerns and we'd appreciate it if you could share your feedback before the discussion period closes. Thank you!

---

### Official Review · Reviewer_8N7P · 2024-11-03

**Soundness:** 3
**Presentation:** 3
**Contribution:** 2
**Rating:** 6
**Confidence:** 4

**Summary:**

The paper proposes a discrete diffusion model that unifies the generative modeling of image and text. To improve the inference efficiency, the paper proposes to denoise image and text with different timesteps. A specific training strategy is proposed correspondingly, which makes image timesteps randomly offsets the text timesteps by a $\delta t$. Experiments are conducted to analyze the unconditional and conditional image-text generation ability, together with image/text retrieval ability, and image-text joint inpainting ability.

**Strengths:**

1. The approach of this paper is novel. Considering that the decoder-only auto-regressive transformers have become the main stream of language modeling, this paper explores diffusion for both image and text, representing a completely different direction.
2. The experiments are conducted comprehensively. In specific settings (relatively small model size and data scale), the proposed method performs better than the AR baseline in unconditional and conditional image-text generation tasks.

**Weaknesses:**

+ About the motivation.
  + The paper says that "generating image tokens autoregressively is slow and wasteful as nearby tokens are highly correlated, and this process results in many unnecessary forward passes through the network". However, from Fig. 4(a) of the paper, AR strikes even better speed-quality trade-off. Moreover, from the conceptual perspective, AR can be more efficient than diffusion models, since AR models benefit from the kv cache during inference, making each forward pass only need to perform the calculation of a single token， while diffusion models have to compute all the tokens in each forward pass.
  + In practice, models usually go through supervised fine-tuning on downstream tasks before actually being put into use. In this way, lots of benefits can be unleashed in the AR paradigm, including generating arbitrary-length sequence, multi-task, multimodal reasoning, and image/text inpainting or editing. However, for diffusion models, due to the token length limitation, tasks involving dynamic sequence length are difficult to implement. Moreover, it is very inflexible when it comes to tuning the pretrained diffusion model into a VQA model for multiple-round conversations.
+ About experiments
  + The paper claims that Perplexity + Entropy is a better indication of the quality of generation results. However, there is no guideline to determine the relative importance of these two metrics when there is trade-off. As a result, it is still unclear whether the proposed method is better than the AR paradigm.
  + From Fig.3, it seems that the proposed method is not more efficient than the AR paradigms.

**Questions:**

1. About image-text retrieval. The paper says that $p(x_{text}|x_{img})$ determines which text is retrieved. For AR model, this is easy to calculate, since the text order is predefined, and the probability can be easily calculated as the cumulative product of the probabilities of all text tokens. However, for the diffusion model, the method for calculating this probability seems unclear. Could the authors clarify this probability calculation process?
2. About details. What is the meaning of the notation $K$ in section 3.3 paragraph 4 "$\mathcal{U}(\frac{N_{min}}{K}, \frac{N_{min}}{K})$"?

---

> ### Author Response · Authors · 2024-11-21
> **Official Comment by Authors (1/3)**
>
> Thank you for your constructive feedback and helpful suggestions. We address your concerns point-by-point below.
>
> **Q2.1: AR is faster than diffusion due to KV caching as seen in Fig 4(a).**
>
> We appreciate the comment. There is indeed a tradeoff between latency (i.e., the time taken to generate a single example), which is lower with diffusion, and maximum throughput (i.e., number of examples decoded per second), which is higher with AR.
>
> AR does indeed achieve a higher throughput than diffusion, due to KV caching$^1$. We use KV caching in AR for all the results we report. Specifically, in Fig. 4(a) we use the maximum batch size that fits in GPU memory (batch size \= 8).
>
> **However, latency plays a more critical role than throughput for many real-time deployment scenarios, or deployment on edge devices, e.g., interactive on-device inference.** Following your insightful comment, we evaluated all models at a batch size of 1, as seen in this figure ([unidisc-diffusion.github.io/#inference_comparisons](https://unidisc-diffusion.github.io/#inference_comparisons)
>  and in Figure 9 of the updated manuscript). From the figure, it can be seen that UniDisc demonstrates comparable generative perplexity, with much higher diversity and lower latency compared to AR. UniDisc’s low latency compared to AR is attributed to its ability to generate multiple tokens at once.
>
> Recent works such as \[1,2\] have shown similar results and found it is possible to trade off latency and throughput between AR and Diffusion by performing diffusion in the innerloop while doing AR in the outerloop. This can allow one to take advantage of both multi token decoding and KV-caching.
>
> We have revised our manuscript with additional details on these tradeoffs (Section A.8).
>
> \----------------------------------------------------------------------------
>
> 1: This tradeoff is dependent on the time-complexity and thus sequence length of attention, and the efficiency gains due to batching. At BS=1, diffusion models can take advantage of the wide parallelization in modern GPUs while AR w/KV caching results in underutilization. As the batch size increases, AR w/KV caching is able to saturate the GPU and achieve higher throughput.
>
> \[1\] B. Chen et. al., “Diffusion Forcing: Next-token Prediction Meets Full-Sequence Diffusion,” 2024, arXiv:2407.01392.
>
> \[2\] T. Wu et al., “AR-Diffusion: Auto-Regressive Diffusion Model for Text Generation,” 2023, arXiv:2305.09515.

---

> ### Author Response · Authors · 2024-11-21
> **Official Comment by Authors (2/3)**
>
> **Q2.2: AR can be supervised fine-tuned for downstream tasks such as inpainting/editing.**
>
> You are absolutely correct, AR models can be finetuned for specific image inpainting and editing tasks, as also shown in prior work \[3\]. However, this requires paired data that is often hard to obtain as it is not naturally occurring in online datasets, e.g., swap a horse to a zebra. Moreover, numerous creative generations—such as those in \[4\] which simultaneously denoises and combines augmented versions of a single image to generate anagrams—do not rely on any paired data, but rather on the innate compositionality of a generative unconditional diffusion model with CFG, which as we have shown, does not work well for AR models.
>
> To satisfy this concern, we run an **additional experiment on multimodal inpainting by fine-tuning the 340M parameter AR model** on a standard set of multimodal datasets (CC12M, Recap-DataComp-1B) while evaluating UniDisc in a zero shot manner—without any fine-tuning. Specifically, for the AR model, we use a linear masking schedule for the prefix sequence, consisting of a partially masked text and image pair and then predict and supervise the clean sequence, doubling the overall sequence length. We evaluate at multiple noise levels, showing the degradation in performance as the original sequence is increasingly masked.
>
> As seen [here](https://unidisc-diffusion.github.io/#quantitative_inpainting_comparison) (Section A.12 of the updated manuscript), UniDisc outperforms the AR model zero-shot, demonstrating the inherent benefits of the training objective.
>
> We also show qualitative results of zero shot editing ([unidisc-diffusion.github.io/#zero_shot_editing](https://unidisc-diffusion.github.io/#zero_shot_editing)) and inpainting ([unidisc-diffusion.github.io/#zero_shot_inpainting](https://unidisc-diffusion.github.io/#zero_shot_inpainting)) from UniDisc, included in Figures 11, 12, and 15\. We will add all these new results in the manuscript.
>
> We summarize several advantages of diffusion versus AR for editing and inpainting tasks below:
>
> 1. Obtaining data for many tasks (e.g., text-conditioned image editing) is highly nontrivial$^1$.
> 2. These tasks are not within the original training distribution and it is often difficult to preserve pre-trained knowledge when fine-tuning for these tasks$^2$.
> 3. Spatial control (e.g., add a dog on the left) cannot be directly controlled and must be learned by the model.
> 4. These approaches are highly inefficient. e.g., editing a small portion of the input requires embedding the entire input sequence and generating an entirely new one, as described above.
>
> \----------------------------------------------------------------------------
>
> 1: For example, \[3\] relies on a diffusion model to generate synthetic data, which itself was trained on data from a base diffusion model. Newer diffusion methods such as \[4\] and ours work zero-shot.
>
> 2: For example, Emu3 has separate fine-tuned models of the same base weights for image generation and conversation.
>
> \[1\] D. Liu et al., “Lumina-mGPT: Illuminate Flexible Photorealistic Text-to-Image Generation with Multimodal Generative Pretraining,” 2024, arXiv:2408.02657.
>
> \[2\] D. Geng, et al., “Visual Anagrams: Generating Multi-View Optical Illusions with Diffusion Models,” 2024, arXiv:2311.17919.
>
> \[3\] L. Yu et al., “Scaling Autoregressive Multi-Modal Models: Pretraining and Instruction Tuning,” Sep. 05, 2023, arXiv: arXiv:2309.02591.
>
> \[4\] Y. Tewel, R. Gal, D. Samuel, Y. Atzmon, L. Wolf, and G. Chechik, “Add-it: Training-Free Object Insertion in Images With Pretrained Diffusion Models,” Nov. 12, 2024, arXiv: arXiv:2411.07232.

---

> ### Author Response · Authors · 2024-11-21
> **Official Comment by Authors (3/3)**
>
> **Q2.3 Dynamic sequence length and tuning for multiple-round QA is difficult with diffusion.**
>
> We agree dynamic sequence generation is less straightforward in discrete diffusion than in AR. We find that we can mitigate some of these limitations in UniDisc.
>
> i) We use RoPE positional embeddings that allow us to dynamically increase the sequence length at test time without requiring fine-tuning. This allows us to dynamically increase the context length similar to AR. Following submission, we demonstrate zero-shot variable resolution generation ([unidisc-diffusion.github.io/#length_extrapolation](https://unidisc-diffusion.github.io/#length_extrapolation) and in Section A.11 of the updated manuscript): **We train UniDisc with 1024 tokens and generate 4096 tokens at test-time**. Similar to AR models, optimal performance is only achieved by fine-tuning with these longer context lengths, and this is an area of future research.
>
> ii) Further, to speed up computation for the text modality, we dynamically ignore the tokens after the end of sequence token. We do so by checking when the model is confident about the prediction of an end of sequence token. Once detected, we remove the tokens after the EOS token, shortening the sequence length and significantly reducing the total number of flops.
>
> **Q2.4: No guideline to determine the relative importance of Generative Perplexity \+ Entropy.**
>
> We add the entropy metric to account for potential cheating by models. Degenerate text samples may obtain low generative perplexity and lead to misleading results unless compensated by entropy. To demonstrate this, we show different text generations and their perplexity score under Chameleon and GPT-2 ([unidisc-diffusion.github.io/#generative_perplexity_metrics](https://unidisc-diffusion.github.io/#generative_perplexity_metrics)). As can be seen, a sentence containing just “AAA” gives lower generative perplexity than a meaningful sentence. This has been highlighted in prior work such as \[3\] which demonstrated that highly repetitive or degenerate samples can trick the generative perplexity metric. We therefore choose to report both metrics.
>
> **Q2.5: Training efficiency**:
>
> We acknowledge this in Lines 382, 398 and in Figure 3 of our paper.
>
> **Q2.6: Details for likelihood calculation for diffusion in Image-text retrieval.**
>
> To perform image-text retrieval with our model, we calculate the ELBO of the sample. Concretely, for $T$ timesteps, we sample a mask from our noise schedule, and compute the ELBO (i.e negative of our loss function). Experimentally, we found that re-weighting the ELBO loss, while giving each timestep equal weightage (i.e., without the $\\frac{\\alpha'\_t}{1 \- \\alpha\_t}$ term) resulted in higher accuracy. We also analyze the results in Figure 5, showing the increase in performance as we vary $T$. We observe that enabling CFG with the given input (e.g., for text retrieval the image is the conditioning), plays a significant role.
>
> For AR retrieval, we can directly calculate the NLL in a single forward pass as is commonly done \[1, 2\]. As noted, for retrieval we train with $p=0.2$ to flip the sequence order, allowing us to place the image before the text and vice-versa.
>
> **Q2.7: Section 3.3 Notation Description**
>
> We apologize for the error. The correct notation is: ${\\delta t}i \\sim  \\mathcal{U}(\\frac{K}{N\_{max}}, \\frac{K}{N\_{min}})$
>
> In other words, we set the max ${\\delta t}i$ based on the observed ratio $K$ and the expected range of sampling steps for text. We have updated the manuscript with this correction.
>
> \----------------------------------------------------------------------------
>
> \[1\] I. Huang et al., “ConMe: Rethinking Evaluation of Compositional Reasoning for Modern VLMs,” Nov. 13, 2024, arXiv: arXiv:2406.08164.
>
> \[2\] J. Li et al., “CoVLM: Composing Visual Entities and Relationships in Large Language Models Via Communicative Decoding,” Nov. 06, 2023, arXiv: arXiv:2311.03354.
>
> \[3\] K. Zheng, Y. Chen, H. Mao, M.-Y. Liu, J. Zhu, and Q. Zhang, “Masked Diffusion Models are Secretly Time-Agnostic Masked Models and Exploit Inaccurate Categorical Sampling,” Oct. 25, 2024, arXiv: arXiv:2409.02908.

---

> > ### Comment · Reviewer_8N7P · 2024-11-26
> >
> > My concerns about the paper are well addressed by the authors. I have decided to raise my score.

---

> ### Author Response · Authors · 2024-11-25
>
> Dear reviewer, we look forward to your response to our rebuttal. The discussion period is nearing its end, we'd appreciate it if you could review it and share your feedback before the discussion period closes. Thank you.

---

### Official Review · Reviewer_fMR4 · 2024-11-06

**Soundness:** 3
**Presentation:** 3
**Contribution:** 3
**Rating:** 6
**Confidence:** 3

**Summary:**

Motivated by the superiority of diffusion models, this work aims to unify text and image generation within a discrete diffusion formulation. The main challenge to unification is the sampling steps for different modalities, e.g., text needs more sampling steps than image. To address this issue, the authors propose to use different time schedules for each modality, and balance the noise schedule accordingly during training. Comprehensive experiments are included to demonstrate how discrete diffusion works for different modalities.

**Strengths:**

- Unifying text and image generation with discrete diffusion sounds given that diffusion process is easier to control, and image diffusion achieved great success recently.
- The writing is clear and easy to follow.
- The motivation to address issue of different time schedules for different modalities is also practical.
Experiments are comprehensive. For example, the authors illustrate the inference efficiency, discriminative ability, fine-tuning from AR models.

**Weaknesses:**

- The main concern is the performance compared to the baselines. The method is good at conditional generation while performing worse on the unconditional generation, with mixed results observed.
- The balance between different modalities seems empirical and handcrafted.

**Questions:**

How do you implement CFG in Chameleon? The model size of Chameleon in experimental comparision. Is it reproduced with the same training set?

---

> ### Author Response · Authors · 2024-11-21
>
> Thank you for your constructive feedback and insights. We address your concerns point-by-point below.
>
> **Q1.1: The performance for UniDisc is worse than Chameleon on Unconditional Generation.**
>
> This is inaccurate: UniDisc gets slightly better results on Unconditional CLIP score (Table 1), while Chameleon gets slightly better results on Unconditional FID (Table 1), which makes them comparable in performance. At the same time, UniDisc’s results for conditional FID and conditional CLIP score are significantly better than that of Chameleon. We believe **conditional generation is the predominant task for generative models in the real-world**, e.g., text-to-image generation, question-answering, image editing, etc. In fact, many papers omit unconditional metrics altogether for this reason \[1,2,3,5,6\], particularly when training on multiple datasets; we include them for completeness.
>
> \
> **Q1.2**  **The data balance between different modalities seems handcrafted.**
>
> Our experiments follow common practices for training multimodal (vision-language) models. Specifically, we use standard web-scraped image-text datasets such as LAION 400M, CC12M and Recap-DataComp-1B. These datasets are representative of the paired image-text content available on the web and have been used in unified image-text models such as Emu3 and BLIP-2 for pre-training.
>
> \
> **Q1.3: How do you implement Classifier-Free Guidance in Chameleon**?
>
> We implement Classifier-Free Guidance (CFG) following standard practice used in \[1,2,3,6\] where we replace the text conditioning during training with a learnable unconditional token with a probability of 0.1, and use this unconditional token at inference.
>
> The CFG results in the paper were obtained with the CFG scale set to 4 for all models, following the optimal value in CM3Leon \[1\]. Since submission, we have run a sweep across CFG scales across all datasets and model sizes, evaluating FID and CLIP scores. We found optimal guidance scale values for AR and UniDisc, which improve performance for both models—but the performance of UniDisc remains higher than the AR model, as you can see at [unidisc-diffusion.github.io/#cfg_ablation](https://unidisc-diffusion.github.io/#cfg_ablation) and in Figure 18 of the updated manuscript.
>
> Further, in Section A.7, we conduct analysis to study why CFG doesn’t significantly improve Chameleon’s performance as it does for UniDisc. CFG enforces the model to go in the direction of the conditional predictions while going in the opposite direction of its unconditional predictions. We find that in autoregressive models, the unconditional predictions start becoming very similar to its conditional predictions after the first token gets decoded. This is because the next token to be predicted is highly correlated to its previous token due to the left-to-right decoding strategy. The high correlation makes it simpler for the unconditional model to predict the next token, thus making the guidance play a smaller role in later steps. In UniDisc this doesn’t happen, as the model predicts tokens in random locations in the sequence, making it tougher for the unconditional predictions to match the conditional predictions. We visualize this trend [here](https://unidisc-diffusion.github.io/#cfg_unidisc_ar_analysis) and in Figure 8 of the updated manuscript by plotting the distance between the unconditional and conditional tokens as more tokens are decoded.
>
> \
> **Q1.4: What is the training data for Chameleon? What is the model size of Chameleon in the experimental comparison?**
>
> We train two versions of Chameleon with equal model parameters to our models, that is,   \~100M and \~300M parameters. Chameleon's \[5\] training set is not publicly available, therefore we use popular publicly available training sets consisting of CC12M, Recap-DataComp-1B, and LAION 400M. We use the same training set—including all tokenization—across both UniDisc and the AR baseline to ensure fair comparisons of different modeling strategies.
>
> \----------------------------------------------------------------------------
> \[1\] L. Yu et al., “Scaling Autoregressive Multi-Modal Models: Pretraining and Instruction Tuning,” 2023, arXiv:2309.02591.
>
> \[2\] J. Yu et al., “Scaling Autoregressive Models for Content-Rich Text-to-Image Generation,” 2022, arXiv:2206.10789.
>
> \[3\] P. Sun et al., “Autoregressive Model Beats Diffusion: Llama for Scalable Image Generation,” 2024, arXiv:2406.06525.
>
> \[4\]: J. Ho and T. Salimans, “Classifier-Free Diffusion Guidance,” 2022, arXiv:2207.12598.
>
> \[5\]: Chameleon Team, “Chameleon: Mixed-Modal Early-Fusion Foundation Models,” 2024, arXiv:2405.09818.
>
> \[6\]: D. Liu et al., “Lumina-mGPT: Illuminate Flexible Photorealistic Text-to-Image Generation with Multimodal Generative Pretraining,” 2024, arXiv:2408.02657.
>
> \[7\] J. Li, D. Li, S. Savarese, and S. Hoi, “BLIP-2: Bootstrapping Language-Image Pre-training with Frozen Image Encoders and Large Language Models,” 2023, arXiv:2301.12597

---

> > ### Comment · Reviewer_fMR4 · 2024-12-02
> > **Thanks for the response**
> >
> > Thanks the authors for the detailed response. In general, this work demonstrates some empirical results for an unified multimodal models. I prefer to maintain my score.

---

> ### Author Response · Authors · 2024-11-25
>
> Dear reviewer, we look forward to your response to our rebuttal. The discussion period is nearing its end, we'd appreciate it if you could review it and share your feedback before the discussion period closes. Thank you.

---

> ### Author Response · Authors · 2024-11-27
>
> Dear reviewer, we are gently checking once again to see if you had an opportunity to look at our rebuttal. We hope to have addressed all your concerns but if you have any more questions that prevent you from accepting the paper, please let us know!

---

> > ### Author Response · Authors · 2024-12-01
> >
> > Dear reviewer, we are once again gently checking to see if you had an opportunity to look at our rebuttal. We hope to have addressed all your concerns and we'd appreciate it if you could share your feedback before the discussion period closes. Thank you!

---

### Author Response · Authors · 2024-11-21

Thank you for the constructive feedback and for thoroughly reading our work. We appreciate that all reviewers recognize the importance (fMR4, 8N7P, JxTt) of our work, highlighting the novel multimodal caching (fMR4, 8N7P, JxTt) and extensive experiments (8N7P, 8LSP). Reviewers highlight that the paper is clearly written and well structured (fMR4, JxTt).

\
Since submission, and guided by the reviewers comments, we have added new experiments which can be summarized as follows:

1. Scaling our experiments to 340M parameters and to larger datasets to verify the scaling laws of diffusion versus AR (fMR4, JxTt)
2. Fine-tuning AR models for inpainting and demonstrating that UniDisc outperforms these models without any fine-tuning (8N7P)
3. Generating variable length image and sequences from UniDisc to demonstrate that the sequence length can vary from what was seen at training time (8N7P)
4. Hyperparameter sweeps of CFG values for all models and scales, improving performance and demonstrating the tradeoff between quality and prompt adherence. Analysis of the limited contribution of CFG in the AR model (fMR4, JxTt)
5. Demonstration of zero-shot image editing capabilities and numerous additional qualitative visualizations
6. Analysis of UniDisc’s generation process by visualizing and segmenting intermediate predictions.

\
These experiments can be found in the updated appendix and at [unidisc-diffusion.github.io](https://unidisc-diffusion.github.io/). Below, we address each reviewer's concerns as well as additional comments.

---

### Meta-Review · Area_Chair_oNQk · 2024-12-21

**Metareview:**

The paper presents UniDisc, an image-text generative model that uses discrete diffusion for encoding and decoding the two disparate modalities jointly. The paper extends prior work on discrete diffusion for text (e.g., Sahoo et al., 2024, Chang et al., 2022, 2023, Austin et al., 2021, etc.) to use image and text modalities. The key innovation in UniDisc is in deriving a method that could allow generation across modalities that have different information densities per token, for which the paper proposes to use different sampling time schedules. Experiments are provided on a variety of tasks and demonstrates advantages against auto-regressive models.

**Additional Comments On Reviewer Discussion:**

The paper received mixed reviews, with three borderline accepts and one borderline reject. The reviewers appreciated the clarity in the presentation, as well as the idea of presenting multimodal discrete diffusion, the use of different time schedules for different modalities, and the extensive experiments comparing to Chameleon on various tasks.

There were some important concerns raised that were extensively discussed with the authors. There are four aspects of the paper that the reviewers criticized on:
1. Limited technical contributions, hand-crafted approach (fMR4), exaggerated claims against UniD3 (JxTt)
2. Lack of experiments against state-of-the-art models (JxTt, 8N7P)
3. Mixed empirical results, especially on conditional and unconditional generation (Table 1), (fMR4, 8N7P)
4. Missing visualizations and architecture diagrams (8LSP)

Authors provided a strong rebuttal addressing some of these concerns. Specifically, for addressing point #2, they presented new results comparing different model sizes and scales, however comparisons to state-of-the-art models will need significant additional compute, and thus could not be done. For addressing #3, authors provided clarification, and pointed to the results using conditional generation that are significant against the prior work, and which is a more prominent task. For point #4, authors provided extensive number of qualitative results and added a diagram depicting the method in the appendix.

A concern that remains in the paper is point #1 on the lack of a significant technical novelty as pointed out by JxTt and fMR4. As noted by JxTt, a prior work UniD3 (ICLR 2023) presents an image-text discrete diffusion model, similar to the presented approach. While, there are minor differences between the exact technicalities, e.g., as noted by the authors, UniD3 uses a single linear schedule for both modalities, while the current paper uses a different schedule, differences in attention, etc., such differences look minor from a technical perspective. If it were important, AC thinks it is inevitable that the paper should make comparisons against UniD3 in a manner that is feasible as without which it is difficult to gauge the merit of the presented innovations. As such, the paper appears to need more work in substantiating its claims empirically or demonstrating state-of-the-art performance, and unfortunately cannot be accepted in its current form.

---

### Decision · Program_Chairs · 2025-01-22

Reject